# Dynamic Influence Tracker: Estimating Sample Influence in SGD-Trained Models across Arbitrary Time Windows

## Abstract

Understanding how training samples affect models improves model interpretability, optimization strategies, and anomaly detection. However, existing methods for estimating sample influence provide only static assessments, rely on restrictive assumptions, and require high computational costs. We propose Dynamic Influence Tracker (DIT), a novel method to estimate time-varying sample influence in models trained with Stochastic Gradient Descent (SGD). DIT enables fine-grained analysis of sample influence within arbitrary time windows during training through a two-phase algorithm. The training phase efficiently captures and stores necessary information about the SGD trajectory, while the inference phase computes the influence of samples on the model within a specified time window. We provide a theoretical error bound for our estimator without assuming convexity, showing its reliability across various learning scenarios. Our experimental results reveal the evolution of sample influence throughout the training process, enhancing understanding of learning dynamics. We show DIT's effectiveness in improving model performance through anomalous sample detection and its potential for advancing curriculum learning.

## 1 Introduction

Deep neural networks, optimized via Stochastic Gradient Descent (SGD) (Bottou, 2010), have achieved remarkable success across various domains. Despite these achievements, it is challenging to estimate the dynamic influence of training samples on the learning process. This understanding is key to enhancing model interpretability, improving optimization strategies, designing effective curriculum learning (Bengio et al., 2009), and enabling early anomaly detection (Chandola et al., 2009).

Existing methods for estimating sample influence, such as influence functions (Koh & Liang, 2017) and SGD-influence (Hara et al., 2019), provide foundational insights but face several limitations: First, existing methods can only estimate the overall impact of samples on the final model. They provide a single, static estimate of influence for the entire training process, failing to capture how the influence evolves throughout different stages of the training process. Second, they often rely on strong assumptions about loss convergence, convexity or model optimality. These conditions are rarely met in modern deep learning environments featuring complex, non-convex loss landscapes. This can lead to inaccurate influence assessments. Third, most methods involve computationally intensive operations, such as retraining the model multiple times (Ghorbani & Zou, 2019) or inverting the Hessian matrix (Koh & Liang, 2017). The high computational costs limit practical applicability and make real-time influence analysis infeasible. Collectively, these limitations obscure the time-varying nature of sample influence, thus restricting the utility and applicability of existing models in real-world scenarios.

To address these challenges, we propose the Dynamic Influence Tracker (DIT) to estimate the time-varying influence of training samples on models trained using SGD. Our method enables to estimate sample influence within arbitrary time windows through a two-phase algorithm. The training phase, executed only once, captures and stores necessary information about the SGD process, particularly focusing on the evolution of the model's parameters over time. The inference phase utilizes the

stored information to compute the influence of selected samples within specified time windows, enabling efficient and flexible analysis.

Compared with existing works, DIT offers the following advantages:

1) **Real-Time and Dynamic Influence Tracking.** DIT provides granular real-time sample influence estimates within arbitrary time windows during model training, capturing dynamic influence fluctuations. Our experiments show that DIT can identify important samples early in training, optimizing the process and enhancing performance in applications such as mislabeled sample detection.

2) **Robustness to Non-Convergence and Non-Convexity.** DIT handles non-convex loss landscapes effectively by utilizing gradient analysis and Hessian-vector approximations without assuming convergence or global optimality. We provide theoretical guarantees on the accuracy of our estimates, showing that estimation errors grow controllably with the training interval, ensuring reliable results even in non-convex settings.

3) **Query-Based Multifaceted Influence Measure.** Our query-based algorithm enables multifaceted analysis of model behavior by projecting parameter changes to specific directions. This approach allows for targeted estimation of how training samples impact loss gradients, predictions, and other model properties, providing a comprehensive understanding of sample influence.

## 2 PRELIMINARIES

Let $\mathcal{Z} = \mathcal{X} \times \mathcal{Y}$ denote the space of observations, where $\mathcal{X} \subseteq \mathbb{R}^d$ is the input space and $\mathcal{Y}$ is the output space. Given a training set $D = \{z_i\}_{i=1}^N$ of i.i.d. observations $z_i = (x_i, y_i) \in \mathcal{Z}$, a model $f : \mathcal{X} \times \Theta \to \mathcal{Y}$ parameterized by $\theta \in \Theta \subseteq \mathbb{R}^p$, and a loss function $\ell : \mathcal{Z} \times \Theta \to \mathbb{R}$, we formulate the learning problem as:

$$\hat{\theta} = \arg\min_{\theta \in \Theta} \frac{1}{N} \sum_{i=1}^N \ell(z_i; \theta). \tag{1}$$

**Definition 1** (Stochastic Gradient Descent (SGD)). Let $g(z; \theta) = \nabla_\theta \ell(z; \theta)$, and SGD starts from $\theta^{[0]}$. The update rule for mini-batch SGD at step $t$ is:

$$\theta^{[t+1]} = \theta^{[t]} - \frac{\eta_t}{|S_t|} \sum_{i \in S_t} g(z_i; \theta^{[t]}), \quad 0 \le t \le T-1, \tag{2}$$

where $S_t \subseteq \{1, ..., N\}$ represents the mini-batch of indices at step $t$, $\eta_t$ is the learning rate at step $t$, and $T$ denotes the total number of SGD steps.

**Definition 2** (Influence Function Koh & Liang (2017)). The influence function measures the impact of removing a single training point $z_j$ on the optimal model parameters $\hat{\theta}$. It is defined as $\hat{\theta}_{-j} - \hat{\theta}$, where $\hat{\theta}_{-j} = \arg\min_\theta \sum_{i=1, i \ne j}^N \ell(z_i; \theta)$. For strongly convex loss functions, it can be approximated as:

$$\hat{\theta}_{-j} - \hat{\theta} \approx -\hat{H}^{-1} \nabla_\theta \ell(z_j; \hat{\theta}), \tag{3}$$

where $\hat{H} = \frac{1}{N} \sum_{z \in D} \nabla^2 \ell(z; \hat{\theta})$ is the Hessian of the loss at the optimal parameters.

**Definition 3** (Counterfactual SGD). The counterfactual SGD process is used to understand the influence of a specific training sample on the learning process by excluding the $j$-th training sample. Starting from $\theta_{-j}^{[0]} = \theta^{[0]}$, the parameters are updated at each step $t$ using:

$$\theta^{[t+1]} = \theta^{[t]} - \frac{\eta_t}{|S_t|} \sum_{i \in S_t \setminus \{j\}} g(z_i; \theta_{-j}^{[t]}), \quad 0 \le t \le T-1. \tag{4}$$

**Definition 4** (SGD-Influence (Hara et al., 2019)). The SGD-influence of training sample $z_j \in D$ within $t$ steps is defined as $\theta_{-j}^{[t]} - \theta^{[t]}$.

While the influence function (Koh & Liang, 2017) provides insights at the optimum, SGD-Influence (Hara et al., 2019) measures the impact of excluding a specific training instance $z_j$ throughout the SGD training process. In the following sections, we will introduce our method for estimating sample influence efficiently for arbitrary time windows during training.

## 3 PARAMETER CHANGE IN TIME WINDOW

### 3.1 PROBLEM FORMULATION

Our goal is to estimate the impact of training samples during an arbitrary time window $[t_1, t_2]$ within the overall training process $[0, T]$, where $0 \le t_1 < t_2 \le T$. We formalize this goal with a counterfactual question: how would the model's parameters change during the interval $[t_1, t_2]$ if a specific sample $z_j$ is not used?

**Definition 5** (Parameter Change in Time Window). For a time window $[t_1, t_2]$ during SGD training, the parameter change estimates the contribution of a training sample $z_j$ as:

$$\Delta\theta_{-j}^{[t_1,t_2]} = (\theta_{-j}^{[t_2]} - \theta_{-j}^{[t_1]}) - (\theta^{[t_2]} - \theta^{[t_1]}), \tag{5}$$

where $(\theta^{[t_2]} - \theta^{[t_1]})$ represents the parameter changes under standard SGD within $[t_1, t_2]$, and $(\theta_{-j}^{[t_2]} - \theta_{-j}^{[t_1]})$ represents the parameter changes over the same interval when excluding sample $z_j$.

For the special case $[0, t]$, starting from the beginning of training, this simplifies to:

$$\Delta\theta_{-j}^{[0,t]} = (\theta_{-j}^{[t]} - \theta_{-j}^{[0]}) - (\theta^{[t]} - \theta^{[0]}) = \theta_{-j}^{[t]} - \theta^{[t]}. \tag{6}$$

For brevity, we denote $\Delta\theta_{-j}^{[0,t]} = \Delta\theta_{-j}^{[t]}$.

### 3.2 ESTIMATION OF PARAMETER CHANGE IN TIME WINDOW

We aim to estimate the parameter change due to the absence of sample $z_j$ over the time window $[t_1, t_2]$, where $0 \le t_1 < t_2 \le T$:

$$\Delta\theta_{-j}^{[t_1,t_2]} = (\theta_{-j}^{[t_2]} - \theta_{-j}^{[t_1]}) - (\theta^{[t_2]} - \theta^{[t_1]}) = (\theta_{-j}^{[t_2]} - \theta^{[t_2]}) - (\theta_{-j}^{[t_1]} - \theta^{[t_1]}). \tag{7}$$

Consider the normal SGD update for step $t$ ($0 \le t \le T - 1$) (including all samples):

$$\theta^{[t+1]} = \theta^{[t]} - \frac{\eta_t}{|S_t|} \sum_{i \in S_t} g(z_i; \theta^{[t]}). \tag{8}$$

Consider the SGD update excluding sample $z_j$:

$$\theta_{-j}^{[t+1]} = \theta_{-j}^{[t]} - \frac{\eta_t}{|S_t|} \sum_{i \in S_t \setminus \{j\}} g(z_i; \theta_{-j}^{[t]}). \tag{9}$$

Calculate the difference between the two updates:

$$\theta_{-j}^{[t+1]} - \theta^{[t+1]} = (\theta_{-j}^{[t]} - \theta^{[t]}) - \frac{\eta_t}{|S_t|} \left( \sum_{i \in S_t \setminus \{j\}} g(z_i; \theta_{-j}^{[t]}) - \sum_{i \in S_t} g(z_i; \theta^{[t]}) \right). \tag{10}$$

Approximate the gradient differences using a first-order Taylor expansion:

$$g(z_i; \theta_{-j}^{[t]}) - g(z_i; \theta^{[t]}) \approx \nabla_\theta g(z_i; \theta^{[t]})^T (\theta_{-j}^{[t]} - \theta^{[t]}), \tag{11}$$

where $\nabla_\theta g(z_i; \theta^{[t]})$ is the gradient of $g(z_i; \theta)$ with respect to $\theta$, evaluated at $\theta^{[t]}$. Define the approximate Hessian matrix $H^{[t]}$ as the average of the outer products of these gradients over the mini-batch:

$$H^{[t]} = \frac{1}{|S_t|} \sum_{i \in S_t} \nabla_\theta g(z_i; \theta^{[t]})^T, \tag{12}$$

Using this definition in Eq.(12) and Eq.(11), we have:

$$\frac{1}{|S_t|} \sum_{i \in S_t} (g(z_i; \theta_{-j}^{[t]}) - g(z_i; \theta^{[t]})) \approx H^{[t]}(\theta_{-j}^{[t]} - \theta^{[t]}). \tag{13}$$

Substituting this approximation into Eq. (10), we have:

$$\theta_{-j}^{[t+1]} - \theta^{[t+1]} \approx (I - \eta_t H^{[t]})(\theta_{-j}^{[t]} - \theta^{[t]}) + \mathbf{1}_{j \in S_t} \frac{\eta_t}{|S_t|} g(z_j; \theta^{[t]}), \tag{14}$$

where $\mathbf{1}_{j \in S_t}$ is an indicator function that equals 1 if $j \in S_t$, otherwise 0. The complete derivation is provided in Appendix A.2.3. Let $Z_t = I - \eta_t H^{[t]}$, $\tilde{\mathbf{1}}_j^{[t]} = \mathbf{1}_{j \in S_t} \frac{\eta_t}{|S_t|} g(z_j; \theta^{[t]})$ and recursively apply this relation over the interval $[t_1, t_2]$:

$$\theta_{-j}^{[t_2]} - \theta^{[t_2]} \approx Z_{t_2-1} Z_{t_2-2} \ldots Z_{t_1}(\theta_{-j}^{[t_1]} - \theta^{[t_1]}) + \sum_{t=t_1}^{t_2-1} Z_{t_2-1} Z_{t_2-2} \ldots Z_{t+1} \tilde{\mathbf{1}}_j^{[t]}. \quad (15)$$

Combining Eq. (7) and Eq. (15), we can get:

$$\Delta\theta_{-j}^{[t_1,t_2]} \approx \left(\prod_{k=t_1}^{t_2-1} Z_k - I\right)(\theta_{-j}^{[t_1]} - \theta^{[t_1]}) + \sum_{t=t_1}^{t_2-1} \left(\prod_{k=t+1}^{t_2-1} Z_k\right) \tilde{\mathbf{1}}_j^{[t]}. \quad (16)$$

We use Eq. (16) for the interval $[0, t_1]$ with $\theta_{-j}^{[0]} = \theta^{[0]}$ to get $(\theta_{-j}^{[t_1]} - \theta^{[t_1]})$:

$$\Delta\theta_{-j}^{[0,t_1]} = \theta_{-j}^{[t_1]} - \theta^{[t_1]} \approx \sum_{t=0}^{t_1-1} \left(\prod_{k=t+1}^{t_1-1} Z_k\right) \tilde{\mathbf{1}}_j^{[t]}. \quad (17)$$

Substituting this Eq. (17) back into Eq. (16), we obtain:

$$\Delta\theta_{-j}^{[t_1,t_2]} \approx \left(\prod_{k=t_1}^{t_2-1} Z_k - I\right)\left(\sum_{t=0}^{t_1-1} \left(\prod_{k=t+1}^{t_1-1} Z_k\right) \tilde{\mathbf{1}}_j^{[t]}\right) + \sum_{t=t_1}^{t_2-1} \left(\prod_{k=t+1}^{t_2-1} Z_k\right) \tilde{\mathbf{1}}_j^{[t]}.$$

We define the estimated parameter change as:

$$\widehat{\Delta\theta}_{-j}^{[t_1,t_2]} = \left(\prod_{k=t_1}^{t_2-1} Z_k - I\right)\left(\sum_{t=0}^{t_1-1} \left(\prod_{k=t+1}^{t_1-1} Z_k\right) \tilde{\mathbf{1}}_j^{[t]}\right) + \sum_{t=t_1}^{t_2-1} \left(\prod_{k=t+1}^{t_2-1} Z_k\right) \tilde{\mathbf{1}}_j^{[t]}.$$

### 3.3 ESTIMATION ERROR ANALYSIS WITHOUT CONVEXITY ASSUMPTIONS

We derive an upper bound on the estimation error $\|\Delta\theta_{-j}^{[t_1,t_2]} - \widehat{\Delta\theta}_{-j}^{[t_1,t_2]}\|$ for our proposed estimator $\widehat{\Delta\theta}_{-j}^{[t_1,t_2]}$ over an arbitrary training interval $[t_1, t_2]$. Under standard non-convex optimization assumptions, we establish the following error bound:

$$\mathbb{E}\left[\left\|\Delta\theta_{-j}^{[t_1,t_2]} - \widehat{\Delta\theta}_{-j}^{[t_1,t_2]}\right\|\right] \leq \frac{\tilde{B}}{M_H}\left(e^{M_H \eta_{\max}(t_2+1)} + e^{M_H \eta_{\max}(t_1+1)} - 2\right), \quad (18)$$

where $M_H$ is the upper bound on the norm of the Hessian matrix $H^{[t]}$, $\eta_{\max}$ is the maximum learning rate, $\tilde{B} = \frac{L_H M^2}{2} + \epsilon_H M$ encapsulates constants related to the Hessian's Lipschitz continuity and approximation error. For detailed derivations and assumptions, see Appendix A.4.

Note that DIT applies to non-converged and non-convex models. The exponential form arises from the recursive nature of error propagation, where each SGD step compounds previous errors multiplicatively. Our analysis is the first to guarantee error bounds for non-converged, non-convex models during arbitrary time windows. The bounds are mathematical guarantees for the worst case, and experimental results show that DIT achieves near-zero errors empirically.

## 4 DYNAMIC INFLUENCE TRACKER: A QUERY-BASED APPROACH

Section 3 discusses how samples affect model parameters, but their impact also extends to loss gradients and predictions. This section introduces DIT, a flexible, query-based method for a comprehensive evaluation of sample effects on model performance.

### 4.1 QUERY-BASED DIT

The core idea of DIT is to project parameter changes onto specific directions in the parameter space, each represe nted by a query vector. This projection enables us to focus on particular aspects of model behavior, reduce the dimensionality of the analysis, and provide interpretable measures of influence. By carefully choosing query vectors, we can investigate how a training sample's influence affects various model aspects.

**Definition 6** (Query-based Dynamic Influence Tracker). Let $q : [0, T] \to \mathbb{R}^p$ be a query function that maps time $t$ to a query vector $q(t) \in \mathbb{R}^p$. The Query-based Dynamic Influence Tracker for a training sample $z_j$ over the time window $[t_1, t_2]$ is defined as:

$$Q_{-j}^{[t_1, t_2]}(q) = \langle q(t_2), \Delta\theta_{-j}^{[t_2]} \rangle - \langle q(t_1), \Delta\theta_{-j}^{[t_1]} \rangle, \tag{19}$$

where $\Delta\theta_{-j}^{[t]} = \theta_{-j}^{[t]} - \theta^{[t]}$ represents the parameter change at time $t$ and $\langle \cdot, \cdot \rangle$ denotes the standard inner product in $\mathbb{R}^p$.

This measure estimates the influence of sample $z_j$ on the model's behavior as projected onto the query directions. For example, by setting $q(t) = \nabla_\theta \ell(z_{\text{test}}; \theta^{[t]})$, we can measure the impact of a training sample on the model's loss for a test point $z_{\text{test}}$:

$$\begin{aligned} Q_{-j}^{[t_1, t_2]}(q) &= \langle \nabla_\theta \ell(z_{\text{test}}; \theta^{[t_2]}), \Delta\theta_{-j}^{[t_2]} \rangle - \langle \nabla_\theta \ell(z_{\text{test}}; \theta^{[t_1]}), \Delta\theta_{-j}^{[t_1]} \rangle \\ &\approx [\ell(z_{\text{test}}; \theta_{-j}^{[t_2]}) - \ell(z_{\text{test}}; \theta^{[t_1]})] - [\ell(z_{\text{test}}; \theta^{[t_2]}) - \ell(z_{\text{test}}; \theta^{[t_1]})]. \end{aligned} \tag{20}$$

Different choices of $q$ enable analysis of various model characteristics. We can set $q = \nabla_\theta f(x_{\text{test}}; \theta^{[t]})$ measures prediction changes, $q = e_i$ (standard basis vector) examines individual parameter importance, and $q = \nabla_\theta \ell(z_j; \theta^{[t]})$ assesses gradient alignments. A detailed analysis of these query vectors is in Appendix A.3.

## 4.2 TRAINING PHASE OF DIT

The training phase captures SGD information in a selectable storage window $W$ (see Algorithm 1).

Empirically, setting $W$ to the first epoch steps achieves better accuracy than baselines while reducing storage from $O(T(|S_t| + p))$ to $O(|W|(|S_t| + p))$, where $T$ is total steps, $|W|$ is window size, $|S_t|$ is batch size, and $p$ is parameter count. The computation complexity remains $O(T \cdot |S_t| \cdot p)$. Periodic compression can further reduce storage overhead.

---

**Algorithm 1** Training Phase of DIT

**Require:** Training dataset $D = \{z_n\}_{n=1}^N$, learning rate $\eta_t$, batch size $|S_t|$, training steps $T$, selectable storage window $W$
**Ensure:** Stored information $A$
1: Initialize model parameters $\theta^{[0]}$
2: Initialize an empty sequence $A$
3: **for** $t = 1$ to $T$ **do**
4:     $S_t = \text{SampleBatch}(D, |S_t|)$
5:     $\theta[t+1] = \theta[t] - \frac{\eta_t}{|S_t|} \sum_{i \in S_t} g(z_i; \theta[t])$
6:     **if** $t \in W$ **then**   $A[t] = \{S_t, \eta_t, \theta[t+1]\}$
7: **end for**
8: **return** $A$

---

## 4.3 INFERENCE PHASE OF DIT

The inference phase computes sample influence on queries for any time window $[t_1, t_2]$ using information stored in window $W$, where $[t_1, t_2]$ is inside $W$.

Algorithm 2 utilizes two key variables, $u_2^{[t]}$ and $u_1^{[t]}$, which propagate $q(t_2)$ and $q(t_1)$ backwards through time while incorporating the $Z_k$ matrices. The algorithm computes $Q$ by summing the inner products of $(u_2^{[t]} - u_1^{[t]})$ with $\tilde{\mathbf{1}}_j^{[t]}$ at each time step. When expanded, this sum precisely matches the structure of $Q_{-j}^{[t_1, t_2]}(q)$ as defined, with the accumulated terms corresponding to $\Delta\theta_{-j}^{[t_2]}$ and $\Delta\theta_{-j}^{[t_1]}$. This approach efficiently computes the influence without explicitly performing large matrix multiplications. See Appendix A.5 for a detailed proof.

---

**Algorithm 2** Inference Phase of DIT

**Require:** Stored information $A$, query function $q$, time window $[t_1, t_2]$, specified sample $z_j$
**Ensure:** Estimated influence $Q$ for sample $z_j$
1: Initialize $Q \leftarrow 0$, $u_1^{[t_2-1]} \leftarrow 0$
2: Initialize $u_2^{[t_2-1]} \leftarrow q(t_2)$
3: **for** $t = t_2 - 1$ **downto** $0$ **do**
4:     **if** $j \in S_t$ **then**
5:         $Q \leftarrow Q + \left\langle (u_2^{[t]} - u_1^{[t]}), \frac{\eta_t}{|S_t|} g(z_j; \theta^{[t]}) \right\rangle$
6:     **end if**
7:     $u_1^{[t-1]} \leftarrow u_1^{[t]} - \eta_t H^{[t]} u_1^{[t]}$
8:     $u_2^{[t-1]} \leftarrow u_2^{[t]} - \eta_t H^{[t]} u_2^{[t]}$
9:     **if** $t = t_1$ **then**   $u_1^{[t-1]} \leftarrow q(t_1)$
10: **end for**
11: **return** $Q$

---

Its time complexity is $O(t_2|S_t|p)$ and space complexity is $O(p)$. DIT avoids the computationally intensive direct computation and storage of the Hessian matrix, which typically requires $O(Tp^2)$ operations. Instead, DIT efficiently computes Hessian-vector products $H^{[t]}u = \nabla_\theta \langle u, g(z; \theta^{[t]}) \rangle$, requiring only $O(|S_t|p)$ operations per iteration. This optimization effectively handles large models and datasets in modern machine-learning contexts.

## 5 EXPERIMENTS

We evaluate DIT through a series of experiments designed to answer the following questions:

- How do training sample influences evolve throughout the learning process?
- How accurately does DIT estimate sample influence compared to existing methods?
- What can we learn by analyzing how influence evolves across various training stages?
- Can time window analysis of sample influence improve practical ML tasks?

### 5.1 EXPERIMENTAL SETUP

We evaluate DIT across diverse datasets and model architectures, comparing it against leading influence estimation methods. Our experimental setup included eight Nvidia RTX A5000 GPUs, each equipped with 24 GB of memory. These were supported by dual Intel Xeon Gold 6342 CPUs running at 2.80GHz with 96 cores in total and 503 GB of RAM. The software environment comprised Ubuntu 22.04.3 LTS (64-bit), PyTorch v2.4.1, CUDA 12.4, and Python 3.11.9. Code and data are available at https://github.com/dynamic-infl-tracker/DIT.

**Datasets** We used four datasets spanning different domains and complexities: Adult (tabular) (Dua & Graff, 2019), 20Newsgroups (text) (Lang, 1995), MNIST (LeCun et al., 2010) and EMNIST (Cohen et al., 2017) (grayscale images). Details are in Appendix A.6.1.

**Models** We used three model architectures of varying complexity: 1) Logistic Regression (LR), a simple linear model serving as a convex baseline; 2) Deep Neural Network (DNN), with two hidden layers using ReLU activations; and 3) Convolutional Neural Network (CNN), with two convolutional layers followed by a fully connected layer. The DNN and CNN represent non-convex scenarios. All models are optimized using the binary cross-entropy loss with logits, which combines sigmoid activation with binary cross-entropy loss for binary classification tasks. Input and output dimensions were adapted to each dataset. Detailed specifications are provided in Appendix A.6.1.

**Comparison Methods** We evaluate DIT against two established methods.

Leave-One-Out (LOO) directly measures the influence of removing a training sample $z_j$ by retraining models. $\Delta\ell_{LOO}(z_j) = \frac{1}{M} \sum_{i=1}^{M} (\ell(z_i, \theta_{-j}) - \ell(z_i, \theta))$, where $z_i \in D_{\text{test}}$, $M$ is the size of the test set $D_{\text{test}} = \{z_i\}_{i=1}^{M}$. While LOO provides a robust ground truth baseline, it is computationally intensive.

Influence Functions (IF) (Koh & Liang, 2017) estimates the influence of removing a training sample $z_j$ on the model's overall loss for a test set $D_{\text{test}}$: $I(z_j, D_{\text{test}}) = -\frac{1}{M} \sum_{i=1}^{M} \nabla_\theta \ell(z_i, \theta)^T H^{-1} \nabla_\theta \ell(z_j, \theta)$, where $H$ is the Hessian of the model's loss at $\theta$.

For DIT, we estimate influence by setting $q(t) = \frac{1}{M} \sum_{i=1}^{M} \nabla_\theta \ell(z_i; \theta^{[t]})$, measuring the impact on test set $D_{\text{test}}$ loss across time window $[t_1, t_2]$: $Q_{-j}^{[t_1, t_2]}(q) \approx \frac{1}{M} \sum_{i=1}^{M} \left[ \ell(z_i; \theta_{-j}^{[t_2]}) - \ell(z_i; \theta_{-j}^{[t_1]}) \right] - \frac{1}{M} \sum_{i=1}^{M} \left[ \ell(z_i; \theta^{[t_2]}) - \ell(z_i; \theta^{[t_1]}) \right]$.

To ensure the reproducibility and robustness of our results, we present them as the mean ± standard deviation calculated over 16 runs, and each initialized with a different random seed.

## 5.2 PATTERNS OF SAMPLE INFLUENCE DYNAMICS

While existing methods typically provide a static estimate of sample influence for the entire training process, our study shows that sample influence on model performance is dynamic and evolves over time. To uncover this, we conducted a preliminary exploration using LOO, as it provides a ground truth assessment of each sample's influence on model performance. Our methodology involved randomly selecting 256 training samples and using LOO to evaluate their loss change at each epoch during model training. For each sample and each epoch, we temporarily removed the sample from the training set, retrained the model for that epoch, and recorded the resulting change in loss. This process generated a time series of sample influences, allowing us to track how the importance of each sample evolved throughout the training process.

As the model converges during training, the loss change decreases with increasing epochs. To identify patterns of sample influence rather than relative influences, we normalized the values within each epoch using StandardScaler. We then used linear regression to analyze trends in influence changes. Figure 1 shows four distinct influence evolution patterns, displaying centroid values for each group. Detailed experimental settings are provided in Appendix A.6.2.

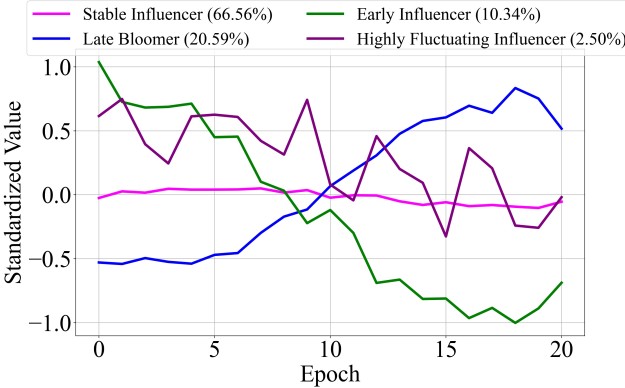

Figure 1: Illustration of influence dynamics patterns for MNIST training using DNN

- **Stable Influencers**: Consistent influence throughout training.
- **Early Influencers**: Significant early impact that diminishes over time.
- **Late Bloomers**: Gain influence as training progresses.
- **Highly Fluctuating Influencers**: Large variations in influence across training.

We further analyzed the pattern distribution across datasets and models, as shown in Table 1.

Table 1: Distribution of influence dynamic patterns across datasets and models (percentage)

| Model | Dataset | Stable Influencer | Early Influencers | Late Bloomers | Highly Fluctuating |
|---|---|---|---|---|---|
| LR | Adult | $64.75 \pm 7.20$ | $11.67 \pm 3.27$ | $20.15 \pm 5.87$ | $3.42 \pm 1.82$ |
| | 20News | $85.94 \pm 5.38$ | $1.17 \pm 1.28$ | $5.57 \pm 1.26$ | $7.32 \pm 4.24$ |
| | MNIST | $80.16 \pm 12.10$ | $0.79 \pm 0.96$ | $10.78 \pm 9.35$ | $8.27 \pm 3.36$ |
| | EMNIST | $75.49 \pm 8.40$ | $0.70 \pm 0.53$ | $13.77 \pm 6.77$ | $10.04 \pm 2.75$ |
| DNN | Adult | $97.91 \pm 2.66$ | $0.313 \pm 1.12$ | $1 \pm 1.55$ | $0.78 \pm 0.89$ |
| | 20News | $79.03 \pm 7.78$ | $8.44 \pm 4.11$ | $11.41 \pm 3.90$ | $1.13 \pm 0.83$ |
| | MNIST | $66.56 \pm 13.26$ | $10.34 \pm 4.65$ | $20.59 \pm 9.44$ | $2.5 \pm 0.93$ |
| | EMNIST | $78.16 \pm 14.48$ | $7.09 \pm 7.678$ | $7.47 \pm 9.87$ | $7.28 \pm 3.55$ |
| CNN | MNIST | $83.76 \pm 19.91$ | $0.34 \pm 0.42$ | $11.74 \pm 16.60$ | $4.15 \pm 3.94$ |
| | EMNIST | $86.50 \pm 7.50$ | $1.87 \pm 5.15$ | $1.59 \pm 3.91$ | $10.03 \pm 2.48$ |

These results show several key insights. 1) All datasets and models show diverse influence patterns, with Stable Influencers dominating but other patterns consistently present. This underscores

the dynamic nature of sample influence throughout the training process. 2) The consistent presence of Early Influencers and Late Bloomers highlights the importance of time-varying analysis in understanding sample influence. DIT's ability to capture these temporal dynamics provides a significant advantage over static influence estimation methods. 3) The varying distributions of influence patterns across different model-dataset combinations show a complex interplay between data characteristics and model architecture. This complexity further emphasizes the necessity of a flexible, query-based approach like DIT, which can adapt to different scenarios and provide targeted insights.

## 5.3 INFLUENCE ESTIMATION ACCURACY

To validate DIT's accuracy in estimating influence, we compared DIT against IF using LOO as ground truth. We employed DIT's full-time window $[0, T]$ for a fair comparison with IF, which can only measure overall sample influence on the final model. To evaluate how closely DIT and IF approximate LOO, we adopted four metrics: Pearson and Spearman correlations for linear and monotonic relationships, respectively, Kendall's tau for ordinal relationships, and Jaccard similarity for the top 30% influencers. Detailed metric descriptions are in Appendix A.6.1.

Table 2: Performance comparison of DIT and IF for Logistic Regression and Deep Neural Network

| Model | Dataset | Pearson | | Spearman | | Kendall's Tau | | Jaccard | |
|---|---|---|---|---|---|---|---|---|---|
| | | DIT | IF | DIT | IF | DIT | IF | DIT | IF |
| LR | Adult | **0.99±0.01** | 0.91±0.04 | **0.99±0.01** | 0.93±0.02 | **0.95±0.01** | 0.79±0.04 | **0.91±0.04** | 0.71±0.06 |
| | 20News | **0.99±0.01** | 0.90±0.13 | **0.99±0.01** | 0.94±0.08 | **0.97±0.01** | 0.84±0.13 | **0.95±0.03** | 0.78±0.16 |
| | MNIST | **0.93±0.10** | 0.76±0.14 | **0.98±0.01** | 0.61±0.22 | **0.95±0.02** | 0.49±0.21 | **0.91±0.05** | 0.48±0.14 |
| DNN | Adult | **0.95±0.02** | 0.88±0.04 | **0.95±0.03** | 0.86±0.04 | **0.83±0.06** | 0.69±0.05 | **0.75±0.08** | 0.56±0.07 |
| | 20News | **0.85±0.07** | 0.77±0.05 | **0.85±0.08** | 0.80±0.06 | **0.71±0.08** | 0.62±0.07 | **0.67±0.08** | 0.55±0.07 |
| | MNIST | **0.90±0.07** | 0.25±0.28 | **0.98±0.01** | 0.26±0.33 | **0.90±0.03** | 0.19±0.24 | **0.85±0.05** | 0.27±0.19 |

Table 2 shows several key findings: First, DIT consistently surpasses IF in accuracy across all datasets, model architectures, and evaluation metrics. Second, DIT's advantage is most significant in complex settings like non-convex DNN and complex MNIST. Third, DIT shows superior robustness and reliability, with lower standard deviations across runs compared to IF.

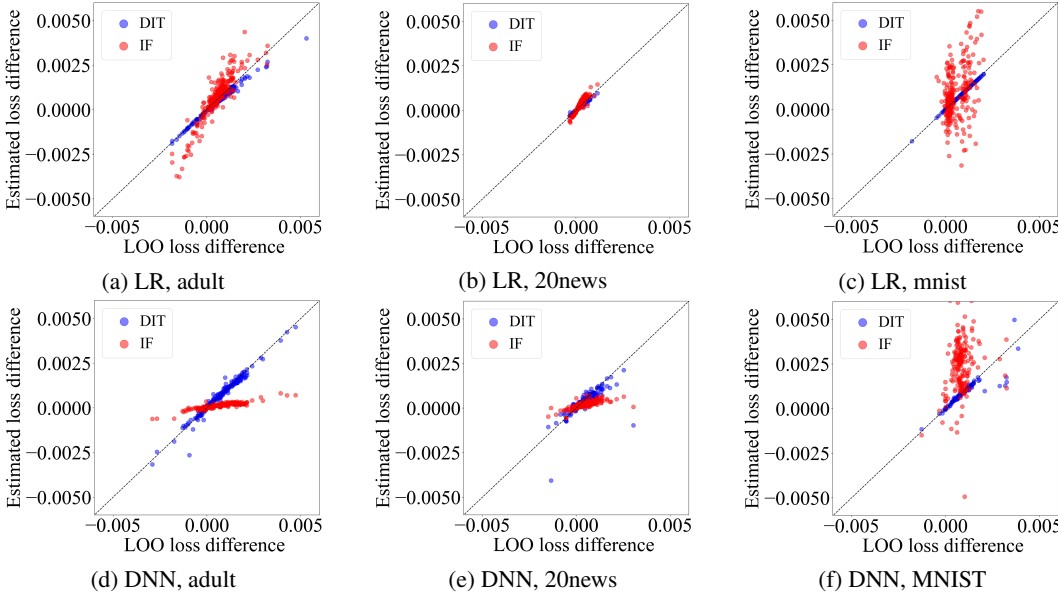

Figure 2: Comparison of influence estimates for DIT and IF vs. LOO ground truth across datasets using LR and DNN. The x-axis represents the ground truth influence values obtained from the LOO method. The y-axis shows DIT (blue) and IF (red) estimates.

These results are visually shown in Figure 2. DIT estimates closely align with the $y = x$ line, indicating superior accuracy to IF, especially with non-convex models and complex datasets.

Furthermore, we analyzed the effectiveness of DIT on samples of different patterns. Due to page limitations, the results are listed in Table 5 in Appendix A.6.2.

### 5.4 INFLUENCE DYNAMICS AND SIMILARITY ACROSS TRAINING STAGES

After validating DIT's accuracy in estimating sample influence, we used it to analyze the similarity of different training stages. The training process was adaptively divided into early, middle, and late stages using change points identified in the overall training loss trajectory. Detailed experimental settings are provided in Appendix A.6.3. Then, we set time windows based on stages and used DIT to compute sample influence within these windows. We then used Kendall's tau correlation to quantify the similarity of influence rankings between stages, with higher values indicating greater stability. Table 3 presents these correlations.

Table 3: Kendall's Tau correlations across training stages

| Model | Dataset | Early-Middle | Early-Late | Middle-Late | Early-Full | Middle-Full | Late-Full |
|---|---|---|---|---|---|---|---|
| | Adult | $0.64 \pm 0.14$ | $0.62 \pm 0.08$ | $0.79 \pm 0.14$ | $0.81 \pm 0.05$ | **$0.82 \pm 0.12$** | $0.79 \pm 0.05$ |
| LR | 20News | $0.79 \pm 0.12$ | $0.78 \pm 0.10$ | $0.79 \pm 0.09$ | **$0.91 \pm 0.02$** | $0.88 \pm 0.10$ | $0.86 \pm 0.12$ |
| | MNIST | $0.43 \pm 0.14$ | $0.15 \pm 0.12$ | $0.35 \pm 0.14$ | $0.71 \pm 0.08$ | **$0.72 \pm 0.09$** | $0.30 \pm 0.14$ |
| | EMNIST | $0.73 \pm 0.04$ | $0.40 \pm 0.16$ | $0.51 \pm 0.18$ | $0.83 \pm 0.03$ | **$0.89 \pm 0.02$** | $0.49 \pm 0.17$ |
| | Adult | $0.61 \pm 0.11$ | $0.41 \pm 0.15$ | $0.70 \pm 0.06$ | $0.7 \pm 0.09$ | **$0.87 \pm 0.04$** | $0.69 \pm 0.08$ |
| DNN | 20news | $0.66 \pm 0.06$ | $0.57 \pm 0.07$ | $0.76 \pm 0.05$ | $0.81 \pm 0.03$ | **$0.82 \pm 0.04$** | $0.76 \pm 0.04$ |
| | MNIST | $0.56 \pm 0.06$ | $0.18 \pm 0.21$ | $0.20 \pm 0.25$ | $0.74 \pm 0.03$ | **$0.81 \pm 0.04$** | $0.20 \pm 0.25$ |
| | EMNIST | $0.60 \pm 0.12$ | $0.40 \pm 0.20$ | $0.59 \pm 0.21$ | $0.69 \pm 0.11$ | **$0.84 \pm 0.07$** | $0.63 \pm 0.17$ |

Table 3 shows several key insights. First, sample influence evolves significantly throughout training, as evidenced by the consistently low correlations between early and late stages (Early-Late column). This challenges the static influence measurement methods and highlights the necessity for time-aware methods like DIT. Second, mid-training influence strongly correlates with full-training influence across all datasets and models. This suggests that influential samples can be identified before convergence. Mid-training analysis may suffice for estimating full-training sample influence, potentially reducing computational costs. These insights have significant implications for data selection and curriculum learning strategies. Third, for a given dataset, the patterns of influence ranking changes at different stages are similar across different model architectures when accounting for standard deviations. This consistency suggests that the influence of samples is largely determined by the inherent dataset rather than being heavily model-dependent.

### 5.5 APPLICATIONS OF DYNAMIC INFLUENCE TRACKER

**Flipped label Sample Detection** To show the practical utility of DIT, we applied it to detecting flipped labels in a binary classification problem using the MNIST dataset (distinguishing between digits '1' and '7'). We randomly selected and flipped labels for 5%, 10%, 15%, and 20% of the training data, corresponding to 12, 25, 38, and 51 samples, respectively. Models were then trained on these partially corrupted datasets. We calculated influence using six methods: full-process DIT, IF, LOO, and epoch-specific DIT (first, middle, and last epochs). For each method, we ranked training samples by their negative influence and evaluated the top-$k$ samples, where $k$ equals the number of deliberately flipped samples. This approach allows us to assess each method's ability to identify mislabeled samples accurately. Table 4 presents results averaged over 16 runs.

First, DIT consistently outperforms IF across all scenarios, often matching or closely approaching the LOO performance. DIT maintains its performance advantage across varying levels of label noise (5% to 20%). Second, the performance gap between DIT and IF widens as model complexity increases (LR < DNN < CNN), highlighting DIT's robustness to non-convexity. Third, later training stages generally yield better detection accuracy, particularly for complex models. As models converge, the influence of mislabeled samples becomes more distinguishable relative to correctly labeled ones. These findings collectively show DIT's effectiveness as a powerful tool for enhancing model robustness and sample quality assessment, particularly in complex, real-world machine learning scenarios.

Table 4: Number of correctly identified flipped samples

| Flipped | Model | IF | Full DIT | LOO | First Epoch DIT | Mid Epoch DIT | Last Epoch DIT |
|---|---|---|---|---|---|---|---|
| | LR | 10.50 ± 0.50 | **10.94 ± 0.90** | **10.94 ± 0.90** | 10.56 ± 1.22 | 10.88 ± 0.78 | 10.88 ± 0.78 |
| 5% | DNN | 2.94 ± 2.01 | 9.06 ± 1.85 | 8.81 ± 1.98 | 8.25 ± 2.33 | 8.88 ± 2.09 | **9.38 ± 1.98** |
| | CNN | 5.88 ± 2.26 | 10.50 ± 1.32 | 10.44 ± 1.32 | 8.75 ± 2.11 | 10.69 ± 1.16 | **11.06 ± 1.32** |
| | LR | 23.44 ± 0.93 | **23.50 ± 1.00** | **23.50 ± 1.00** | 22.56 ± 1.54 | **23.50 ± 1.06** | 23.38 ± 1.00 |
| 10% | DNN | 7.50 ± 3.34 | 20.75 ± 3.01 | 19.94 ± 3.77 | 20.31 ± 2.78 | 20.50 ± 3.22 | **21.31 ± 3.77** |
| | CNN | 15.00 ± 2.83 | 21.81 ± 3.11 | 21.75 ± 3.11 | 18.44 ± 4.37 | 22.19 ± 2.81 | **23.56 ± 3.11** |
| | LR | **36.06 ± 0.97** | **36.06 ± 1.14** | **36.06 ± 1.14** | 35.38 ± 1.62 | 35.69 ± 1.69 | 35.13 ± 1.14 |
| 15% | DNN | 12.63 ± 4.62 | 32.81 ± 3.47 | 32.50 ± 3.72 | 32.19 ± 3.40 | 32.56 ± 3.61 | **33.31 ± 3.72** |
| | CNN | 23.44 ± 4.68 | 34.19 ± 4.17 | 34.19 ± 4.17 | 29.75 ± 5.93 | 34.56 ± 3.98 | **36.31 ± 4.17** |
| | LR | 48.63 ± 1.11 | **48.69 ± 1.16** | **48.69 ± 1.16** | 47.94 ± 1.52 | 46.56 ± 3.12 | 42.94 ± 1.16 |
| 20% | DNN | 22.31 ± 6.14 | 45.31 ± 3.29 | 43.94 ± 5.20 | 44.13 ± 3.64 | 45.19 ± 3.30 | **45.56 ± 5.20** |
| | CNN | 31.00 ± 5.79 | 46.19 ± 4.33 | 46.25 ± 4.35 | 41.50 ± 7.66 | 47.13 ± 3.35 | **48.69 ± 4.35** |

# 6 RELATED WORKS

Estimating the influence of individual training samples on machine learning models is important for optimization and interpretability. While the Leave-One-Out (LOO) method is straightforward, it's computationally prohibitive for large datasets or complex models. Influence functions (Koh & Liang, 2017) offer a more feasible alternative, estimating the impact of removing a single training sample on model performance at convergence. However, their effectiveness is limited in non-convex scenarios common in deep learning (Basu et al., 2021). Recent extensions (Guo et al., 2021; Schioppa et al., 2022; Choe et al., 2024) still provide static, full-process influence measures, failing to capture dynamic sample influence during training.

Shapley Value-based approaches (Ghorbani & Zou, 2019) provide a robust, equitable valuation of individual sample contributions by considering all possible subsets of training data. Efficient approximation algorithms (Jia et al., 2019; 2021; Xu et al., 2021) and domain-specific extensions (Schoch et al., 2022; Sun et al., 2023; Fan et al., 2022) have improved scalability, but remain computationally expensive for large-scale problems.

Data cleansing and pruning focus on removing noisy or irrelevant data. SGD-influence (Hara et al., 2019) analyzes the gradient descent process and estimates sample influence across the entire training trajectory. Our proposed DIT extends this approach, enabling influence estimation within arbitrary time windows during training, providing more flexible error bound analysis and detailed experimental evaluation. Forgetting events (Toneva et al., 2018) and early-training scores (Paul et al., 2021) enable efficient data pruning. MOSO (Tan et al., 2024) identifies less informative samples via gradient deviations, and YOCO (He et al., 2023) enables flexible resizing of condensed datasets.

Despite these advancements, analyzing sample influence within arbitrary time windows during training remains a challenge. DIT addresses this gap by providing a flexible, computationally efficient method for fine-grained influence tracking without relying on strong convexity assumptions. It enables multidimensional influence measurement with a single training process, offering a comprehensive understanding of sample importance throughout the learning trajectory.

# 7 CONCLUSION

This paper introduces Dynamic Influence Tracker (DIT), a novel approach for fine-grained estimation of individual training sample influence within arbitrary time windows in SGD-trained models. Our method's query-based design enables multifaceted analysis of sample influence on various aspects of model performance effectively. Our theoretical analysis provides error bounds without assuming convexity. Extensive experimental results reveal patterns in influence dynamics and show that DIT consistently outperforms existing methods in influence estimation accuracy, particularly for complex models and datasets.

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

# A  APPENDIX

## A.1  PATTERN-SPECIFIC INFLUENCE ESTIMATION

Our experimental analysis in Section 5.2 revealed that sample influence exhibits distinct temporal patterns throughout the training process, categorized as Stable Influencers, Early Influencers, Late Bloomers, and Highly Fluctuating Influencers. To rigorously evaluate DIT's effectiveness across these diverse influence patterns, we conducted a pattern-specific performance analysis comparing DIT against IF with LOO as ground truth using the MNIST dataset with DNN architecture. Table 5 presents the comparative results across multiple metrics. Results show mean±std across 16 runs.

Table 5: Pattern-specific performance comparison between DIT and IF using MNIST-DNN.

| Sample Pattern | Pearson | | Spearman | | Kendall's Tau | | Jaccard | |
| --- | --- | --- | --- | --- | --- | --- | --- | --- |
| | DIT | IF | DIT | IF | DIT | IF | DIT | IF |
| Stable Influencers | **0.95±0.03** | 0.23±0.39 | **0.96±0.03** | 0.16±0.40 | **0.87±0.05** | 0.13±0.28 | **0.82±0.12** | 0.26±0.16 |
| Early Influencers | **0.94±0.04** | 0.35±0.29 | **0.98±0.01** | 0.35±0.30 | **0.92±0.03** | 0.26±0.23 | **0.89±0.07** | 0.29±0.20 |
| Late Bloomers | **0.98±0.02** | 0.23±0.46 | **0.98±0.02** | 0.19±0.38 | **0.90±0.05** | 0.15±0.27 | **0.85±0.10** | 0.27±0.21 |
| Highly Fluctuating | **0.76±0.18** | -0.10±0.54 | **0.72±0.18** | -0.08±0.48 | **0.63±0.21** | -0.09±0.40 | **0.52±0.34** | 0.15±0.17 |

The pattern-specific analysis reveals several key insights:

1) **Substantial Performance Gap:** DIT shows remarkable improvements over IF across all patterns. For Stable Influencers, DIT achieves a 4.1× improvement (Pearson: 0.95±0.03 vs 0.23±0.39), for Late Bloomers, DIT shows a 4.3× improvement (0.98±0.02 vs 0.23±0.46), and for Early Influencers, DIT maintains a 2.7× advantage (0.94±0.04 vs 0.35±0.29). Even in the challenging case of Highly Fluctuating samples, DIT maintains positive correlations (0.76±0.18) while IF shows negative correlations (-0.10±0.54).

2) **Consistency and Stability:** DIT shows remarkable stability in its performance metrics. Linear correlations (Pearson) and rank-based correlations (Spearman, Kendall's Tau) show strong agreement, with variations typically within ±0.05. The standard deviations for DIT are consistently lower than IF, indicating significantly more reliable estimates.

3) **Pattern-Specific Excellence:** DIT excels at capturing influence patterns across different temporal stages. The method achieves near-perfect correlations ($> 0.94$) for Stable, Early, and Late patterns, with a particularly strong performance for Late Bloomers (0.98±0.02). This shows DIT's unique capability to adapt to varying temporal dynamics throughout the training process.

4) **Robustness to Volatility:** Even under challenging conditions with Highly Fluctuating samples, DIT maintains meaningful positive correlations (Pearson: 0.76±0.18). This contrasts sharply with IF's negative correlations (-0.10±0.54), highlighting DIT's robust performance even in volatile scenarios. While performance shows some expected degradation compared to stable patterns, DIT continues to provide reliable influence estimates.

These comprehensive results show DIT's effectiveness in handling diverse influence patterns while maintaining high estimation accuracy. The method's robust performance across both architectural complexity and temporal influence patterns, particularly in challenging scenarios involving non-convex models and fluctuating influences, establishes its practical utility for real-world deep learning applications. The consistent outperformance of IF across all patterns and metrics further validates DIT's advantages in capturing dynamic sample influence.

## A.2 DISCUSSIONS

### A.2.1 KEY FINDINGS AND IMPLICATIONS

Our investigation of Dynamic Influence Tracker (DIT) reveals several important insights about the nature of sample influence in deep learning. We first present a comprehensive comparison of different sample influence analysis methods:

Table 6: Comprehensive Comparison of Different Sample Influence Analysis Methods

| Aspect | Full-DIT | First-Epoch DIT | Influence Functions | Leave-One-Out |
|---|---|---|---|---|
| Space Complexity | $O(T(|S_t| + p))$ | $O(E(|S_t| + p))$ | $O(p^2)$ | $O(p)$ |
| Time Complexity | $O(T|S_t|p)$ | $O(E|S_t|p)$ | $O(p^3 + Np^2)$ | $O(NT|S_t|p)$ |
| Robustness to Non-convergence | Yes | Yes | No | No |
| Robustness to Non-convexity | Yes | Yes | No | Yes |
| Robustness to Global Optimality | Yes | Yes | No | Yes |
| Adaptability | Flexible | Flexible | Static | Static |

$T$ = total steps, $E$ = steps per epoch, $p$ = parameters, $|S_t|$ = batch size

This comparison highlights several key findings:

1) Our results show that sample influence is not static but evolves significantly throughout the training process. The identification of four distinct influence patterns (Stable Influencers, Early Influencers, Late Bloomers, and Highly Fluctuating Influencers) challenges the traditional static view of sample importance.

2) The strong correlation between mid-training and full-training influence measures suggests that influential samples can be identified well before model convergence. This finding has practical implications for efficient training protocols and early intervention strategies.

3) The consistency of influence patterns across different model architectures for the same dataset suggests that sample influence is more intrinsically tied to data characteristics than model architecture.

### A.2.2 APPLICATIONS IN MODERN DEEP LEARNING

There are two particularly promising applications of DIT: large language model fine-tuning and curriculum learning.

In the context of large pre-trained models, DIT addresses several fundamental challenges that have previously limited influence analysis methods. The method's assumption-free nature makes it particularly suitable for fine-tuning scenarios, as it requires no constraints on global optimality or convergence. By treating the pre-trained model's parameters as the initial state $\theta_0$, DIT naturally integrates with existing fine-tuning workflows without requiring complete retraining cycles.

Moreover, the scalability of DIT's model-agnostic design presents a significant advantage for large-scale applications. Whether applied to full model fine-tuning or specific architectural components like adapters, the query-based approach efficiently tracks parameter influence while maintaining computational feasibility. This scalability is further enhanced by DIT's shown ability to identify influential samples in early training stages, enabling effective analysis of large language models while minimizing storage requirements through targeted early-stage tracking.

DIT also offers novel approaches to curriculum learning by providing data-driven methods for sample ordering and difficulty assessment. The identification of distinct influence patterns naturally informs curriculum design: Early Influencers serve as optimal starting points for initial training stages, while Late Bloomers naturally align with curriculum progression. Stable Influencers provide consistent anchoring points across different training phases, enabling automatic difficulty assessment without relying on manual labeling techniques.

This approach to curriculum learning is supported by our empirical findings on cross-stage influence correlations. The strong correlation between early and full training influence enables reliable early identification of important samples. Simultaneously, the observed low correlation between early and late stages provides empirical support for the necessity of progressive learning approaches. These

relationships establish a robust theoretical foundation for dynamic curriculum design, offering a data-driven framework for optimizing training trajectories.

### A.2.3 DETAILED DERIVATION OF PARAMETER CHANGE ESTIMATION

We start from Eq.(10), which establishes the relationship:

$$\theta_{-j}^{[t+1]} - \theta^{[t+1]} = (\theta_{-j}^{[t]} - \theta^{[t]}) - \frac{\eta_t}{|S_t|}(\sum_{i \in S_t \setminus \{j\}} g(z_i; \theta_{-j}^{[t]}) - \sum_{i \in S_t} g(z_i; \theta^{[t]})) \tag{21}$$

$$= (\theta_{-j}^{[t]} - \theta^{[t]}) - \frac{\eta_t}{|S_t|}(\sum_{i \in S_t \setminus \{j\}} g(z_i; \theta_{-j}^{[t]}) - \sum_{i \in S_t \setminus \{j\}} g(z_i; \theta^{[t]}) - \mathbf{1}_{j \in S_t} g(z_i; \theta^{[t]})) \tag{22}$$

$$= (\theta_{-j}^{[t]} - \theta^{[t]}) - \frac{\eta_t}{|S_t|}(\sum_{i \in S_t \setminus \{j\}} g(z_i; \theta_{-j}^{[t]}) - \sum_{i \in S_t \setminus \{j\}} g(z_i; \theta^{[t]})) + \frac{\eta_t}{|S_t|}\mathbf{1}_{j \in S_t} g(z_i; \theta^{[t]}) \tag{23}$$

$$= (\theta_{-j}^{[t]} - \theta^{[t]}) - \frac{\eta_t}{|S_t|}\sum_{i \in S_t \setminus \{j\}}(g(z_i; \theta_{-j}^{[t]}) - g(z_i; \theta^{[t]})) + \frac{\eta_t}{|S_t|}\mathbf{1}_{j \in S_t} g(z_i; \theta^{[t]}), \tag{24}$$

where $\mathbf{1}_{j \in S_t}$ is an indicator function that equals 1 if $j \in S_t$, otherwise 0.

Using Eq.(12), we have:

$$\sum_{i \in S_t \setminus \{j\}} (g(z_i; \theta_{-j}^{[t]}) - g(z_i; \theta^{[t]})) \approx \sum_{i \in S_t \setminus \{j\}} \nabla_\theta g(z_i; \theta^{[t]})^T (\theta_{-j}^{[t]} - \theta^{[t]}), \tag{25}$$

Following Eq.(11) and Assumption (A4) detailed in Appendix A.4, we have:

$$\sum_{i \in S_t \setminus \{j\}} \nabla_\theta g(z_i; \theta^{[t]})^T (\theta_{-j}^{[t]} - \theta^{[t]}) = |S_t| H_{-j}^{[t]} (\theta_{-j}^{[t]} - \theta^{[t]}) \approx |S_t| H^{[t]} (\theta_{-j}^{[t]} - \theta^{[t]}). \tag{26}$$

Combining Eq.(25) and Eq.(26), we have:

$$\sum_{i \in S_t \setminus \{j\}} (g(z_i; \theta_{-j}^{[t]}) - g(z_i; \theta^{[t]})) \approx |S_t| H^{[t]} (\theta_{-j}^{[t]} - \theta^{[t]}). \tag{27}$$

Applying Eq. (27) to Eq. (21), we have the final result:

$$\theta_{-j}^{[t+1]} - \theta^{[t+1]} = (\theta_{-j}^{[t]} - \theta^{[t]}) - \frac{\eta_t}{|S_t|}\sum_{i \in S_t \setminus \{j\}}(g(z_i; \theta_{-j}^{[t]}) - g(z_i; \theta^{[t]})) + \frac{\eta_t}{|S_t|}\mathbf{1}_{j \in S_t} g(z_i; \theta^{[t]})$$

$$\tag{28}$$

$$\approx (\theta_{-j}^{[t]} - \theta^{[t]}) - \frac{\eta_t}{|S_t|}(|S_t| H^{[t]} (\theta_{-j}^{[t]} - \theta^{[t]})) + \frac{\eta_t}{|S_t|}\mathbf{1}_{j \in S_t} g(z_i; \theta^{[t]}) \tag{29}$$

$$= (\theta_{-j}^{[t]} - \theta^{[t]}) - \eta_t H^{[t]} (\theta_{-j}^{[t]} - \theta^{[t]}) + \frac{\eta_t}{|S_t|}\mathbf{1}_{j \in S_t} g(z_i; \theta^{[t]}) \tag{30}$$

$$= (I - \eta_t H^{[t]})(\theta_{-j}^{[t]} - \theta^{[t]}) + \frac{\eta_t}{|S_t|}\mathbf{1}_{j \in S_t} g(z_i; \theta^{[t]}). \tag{31}$$

This derivation confirms the correctness of Eq. (14), including the last term.

### A.3 DIT Toolkit

The flexibility of query-based DIT allows for its application to a wide range of machine learning challenges. In this section, we provide a toolkit of query vectors that enables targeted investigations into critical aspects of model behavior, including gradient value, prediction changes, feature importance, and parameter importance.

#### A.3.1 DIT for Loss Value

**Theorem 7** (DIT for Loss Value). Given a loss function $\ell(z; \theta)$, a time window $[t_1, t_2]$, a training sample $z_j$, and a test sample $z_{\text{test}}$, the Dynamic Influence Tracker with query function $q(t) = (\nabla_\theta \ell(z_{\text{test}}; \theta^{[t]})$ can be approximated as:

$$Q_{-j}^{[t_1, t_2]}(q) \approx [\ell(z_{\text{test}}; \theta_{-j}^{[t_2]}) - \ell(z_{\text{test}}; \theta_{-j}^{[t_1]})] - [\ell(z_{\text{test}}; \theta^{[t_2]}) - \ell(z_{\text{test}}; \theta^{[t_1]})], \tag{32}$$

where $\theta_{-j}^{[t]}$ denotes the model parameters at time $t$ when trained without sample $z_j$, and $\theta^{[t]}$ denotes the parameters when trained with all samples.

*Proof.* We begin with the definition of the Query-Based Dynamic Influence Tracker:

$$Q_{-j}^{[t_1, t_2]}(q) = \left\langle q(t_2), \Delta\theta_{-j}^{[t_2]} \right\rangle - \left\langle q(t_1), \Delta\theta_{-j}^{[t_1]} \right\rangle \tag{33}$$

where $\Delta\theta_{-j}^{[t]} = \theta_{-j}^{[t]} - \theta^{[t]}$.

Substituting $q(t) = \nabla_\theta \ell(z_{\text{test}}; \theta^{[t]})$ into Eq. (33):

$$Q_{-j}^{[t_1, t_2]}(q) = \left\langle \nabla_\theta \ell(z_{\text{test}}; \theta^{[t_2]}), \theta_{-j}^{[t_2]} - \theta^{[t_2]} \right\rangle - \left\langle \nabla_\theta \ell(z_{\text{test}}; \theta^{[t_1]}), \theta_{-j}^{[t_1]} - \theta^{[t_1]} \right\rangle. \tag{34}$$

Apply the first-order Taylor expansion of $\ell(z_{\text{test}}; \theta)$ around $\theta^{[t_2]}$ and $\theta^{[t_1]}$:

$$\ell(z_{\text{test}}; \theta_{-j}^{[t_2]}) \approx \ell(z_{\text{test}}; \theta^{[t_2]}) + \langle \nabla_\theta \ell(z_{\text{test}}; \theta^{[t_2]}), \theta_{-j}^{[t_2]} - \theta^{[t_2]} \rangle \tag{35}$$

$$\ell(z_{\text{test}}; \theta_{-j}^{[t_1]}) \approx \ell(z_{\text{test}}; \theta^{[t_1]}) + \langle \nabla_\theta \ell(z_{\text{test}}; \theta^{[t_1]}), \theta_{-j}^{[t_1]} - \theta^{[t_1]} \rangle \tag{36}$$

Rearranging Eq. (35) and Eq. (36):

$$\langle \nabla_\theta \ell(z_{\text{test}}; \theta^{[t_2]}), \theta_{-j}^{[t_2]} - \theta^{[t_2]} \rangle \approx \ell(z_{\text{test}}; \theta_{-j}^{[t_2]}) - \ell(z_{\text{test}}; \theta^{[t_2]}) \tag{37}$$

$$\langle \nabla_\theta \ell(z_{\text{test}}; \theta^{[t_1]}), \theta_{-j}^{[t_1]} - \theta^{[t_1]} \rangle \approx \ell(z_{\text{test}}; \theta_{-j}^{[t_1]}) - \ell(z_{\text{test}}; \theta^{[t_1]}) \tag{38}$$

Substituting these approximations back into Eq. (34):

$$Q_{-j}^{[t_1, t_2]}(q) \approx [\ell(z_{\text{test}}; \theta_{-j}^{[t_2]}) - \ell(z_{\text{test}}; \theta^{[t_2]})] - [\ell(z_{\text{test}}; \theta_{-j}^{[t_1]}) - \ell(z_{\text{test}}; \theta^{[t_1]})] \tag{39}$$

$$= [\ell(z_{\text{test}}; \theta_{-j}^{[t_2]}) - \ell(z_{\text{test}}; \theta_{-j}^{[t_1]})] - [\ell(z_{\text{test}}; \theta^{[t_2]}) - \ell(z_{\text{test}}; \theta^{[t_1]})] \tag{40}$$

This completes the proof of Theorem 7. □

This theorem provides a foundation for understanding how individual training samples affect the model's loss on specific test points over time. The right-hand side of Eq. (32) represents the difference between the loss changes with and without sample $z_j$, offering a direct measure of the sample's influence on model performance.

**Extension to Test Sets:** We can extend this concept to consider an entire test set $D_{\text{test}} = \{z_1, \ldots, z_M\}$. Define the query function as:

$$q(t) = \frac{1}{M} \sum_{i=1}^{M} \nabla_\theta \ell(z_i; \theta^{[t]}), \quad z_i \in D_{\text{test}}. \tag{41}$$

With this choice, the DIT approximates the change in average test loss:

$$Q_{-j}^{[t_1, t_2]}(q) \approx \frac{1}{M} \sum_{i=1}^{M} \left[ \ell(z_i; \theta_{-j}^{[t_2]}) - \ell(z_i; \theta_{-j}^{[t_1]}) \right] - \frac{1}{M} \sum_{i=1}^{M} \left[ \ell(z_i; \theta^{[t_2]}) - \ell(z_i; \theta^{[t_1]}) \right] \tag{42}$$

$$= \left[ \mathcal{L}_{\text{test}}(\theta_{-j}^{[t_2]}) - \mathcal{L}_{\text{test}}(\theta_{-j}^{[t_1]}) \right] - \left[ \mathcal{L}_{\text{test}}(\theta^{[t_2]}) - \mathcal{L}_{\text{test}}(\theta^{[t_1]}) \right],$$

where $\mathcal{L}_{\text{test}}(\theta^{[t]}) = \frac{1}{M} \sum_{i=1}^{M} \ell(z_i; \theta^{[t]})$ is the average test loss.

### A.3.2 DIT FOR PREDICTION CHANGES

**Theorem 8** (DIT for Prediction Changes). Given a model function $f(x; \theta)$, a time window $[t_1, t_2]$, a training sample $z_j$, and a test input $x_{\text{test}}$, the Dynamic Influence Tracker with query function $q(t) = \nabla_\theta f(x_{\text{test}}; \theta^{[t]})$ can be approximated as:

$$Q_{-j}^{[t_1, t_2]}(q) \approx \left[ f(x_{\text{test}}; \theta_{-j}^{[t_2]}) - f(x_{\text{test}}; \theta_{-j}^{[t_1]}) \right] - \left[ f(x_{\text{test}}; \theta^{[t_2]}) - f(x_{\text{test}}; \theta^{[t_1]}) \right], \quad (43)$$

where $\theta_{-j}^{[t]}$ denotes the model parameters at time $t$ when trained without sample $z_j$, and $\theta^{[t]}$ denotes the parameters when trained with all samples.

*Proof.* We begin with the definition of the Query-Based Dynamic Influence Tracker:

$$Q_{-j}^{[t_1, t_2]}(q) = \left\langle q(t_2), \Delta\theta_{-j}^{[t_2]} \right\rangle - \left\langle q(t_1), \Delta\theta_{-j}^{[t_1]} \right\rangle \quad (44)$$

where $\Delta\theta_{-j}^{[t]} = \theta_{-j}^{[t]} - \theta^{[t]}$.

Substituting $q(t) = \nabla_\theta f(z_{\text{test}}; \theta^{[t]})$ into Eq. (44):

$$Q_{-j}^{[t_1, t_2]}(q) = \left\langle \nabla_\theta f(z_{\text{test}}; \theta^{[t_2]}), \theta_{-j}^{[t_2]} - \theta^{[t_2]} \right\rangle - \left\langle \nabla_\theta f(z_{\text{test}}; \theta^{[t_1]}), \theta_{-j}^{[t_1]} - \theta^{[t_1]} \right\rangle. \quad (45)$$

We apply the first-order Taylor approximation of the model function around $\theta^{[t_2]}$ and $\theta^{[t_1]}$:

$$f(x_{\text{test}}; \theta_{-j}^{[t_2]}) \approx f(x_{\text{test}}; \theta^{[t_2]}) + \langle \nabla_\theta f(x_{\text{test}}; \theta^{[t_2]}), \theta_{-j}^{[t_2]} - \theta^{[t_2]} \rangle \quad (46)$$

$$f(x_{\text{test}}; \theta_{-j}^{[t_1]}) \approx f(x_{\text{test}}; \theta^{[t_1]}) + \langle \nabla_\theta f(x_{\text{test}}; \theta^{[t_1]}), \theta_{-j}^{[t_1]} - \theta^{[t_1]} \rangle \quad (47)$$

Rearranging these equations:

$$\langle \nabla_\theta f(x_{\text{test}}; \theta^{[t_2]}), \theta_{-j}^{[t_2]} - \theta^{[t_2]} \rangle \approx f(x_{\text{test}}; \theta_{-j}^{[t_2]}) - f(x_{\text{test}}; \theta^{[t_2]}) \quad (48)$$

$$\langle \nabla_\theta f(x_{\text{test}}; \theta^{[t_1]}), \theta_{-j}^{[t_1]} - \theta^{[t_1]} \rangle \approx f(x_{\text{test}}; \theta_{-j}^{[t_1]}) - f(x_{\text{test}}; \theta^{[t_1]}) \quad (49)$$

Substituting these approximations back into Eq. (45):

$$Q_{-j}^{[t_1, t_2]}(q) \approx [f(x_{\text{test}}; \theta_{-j}^{[t_2]}) - f(x_{\text{test}}; \theta^{[t_2]})] - [f(x_{\text{test}}; \theta_{-j}^{[t_1]}) - f(x_{\text{test}}; \theta^{[t_1]})] \quad (50)$$

$$= [f(x_{\text{test}}; \theta_{-j}^{[t_2]}) - f(x_{\text{test}}; \theta_{-j}^{[t_1]})] - [f(x_{\text{test}}; \theta^{[t_2]}) - f(x_{\text{test}}; \theta^{[t_1]})] \quad (51)$$

This completes the proof of Theorem 8. □

This theorem provides a formal justification for using DIT to analyze how excluding sample $z_j$ influences the model's predictions on a test input $x_{\text{test}}$ over the interval $[t_1, t_2]$. Compared to Theorem 7, which focuses on the loss value, Theorem 8 focuses on specific model outputs. It enables the identification of influential training samples for specific predictions, aids in understanding model behavior on particular inputs, and can help detect potential outliers or mislabeled data.

### A.3.3 DIT FOR FEATURE IMPORTANCE

**Theorem 9** (DIT for Feature Importance). Given a loss function $\ell(z; \theta)$, a training sample $z = (x, y)$, and a test sample $z_{\text{test}} = (x_{\text{test}}, y_{\text{test}})$, the Dynamic Influence Tracker for Feature Importance with query function $q(t) = \nabla_x \nabla_\theta \ell(z_{\text{test}}; \theta^{[t]})$ can be approximated as:

$$Q_{-j}^{[t_1, t_2]}(q) \approx [\nabla_x \ell(z_{\text{test}}; \theta_{-j}^{[t_2]}) - \nabla_x \ell(z_{\text{test}}; \theta_{-j}^{[t_1]})] - [\nabla_x \ell(z_{\text{test}}; \theta^{[t_2]}) - \nabla_x \ell(z_{\text{test}}; \theta^{[t_1]})], \quad (52)$$

where $\theta_{-j}^{[t]}$ denotes the model parameters at time $t$ when trained without sample $z_j$, and $\theta^{[t]}$ denotes the parameters when trained with all samples.

*Proof.* We start with the definition of the Query-Based Dynamic Influence Tracker:

$$Q_{-j}^{[t_1, t_2]}(q) = \left\langle q(t_2), \Delta\theta_{-j}^{[t_2]} \right\rangle - \left\langle q(t_1), \Delta\theta_{-j}^{[t_1]} \right\rangle, \quad (53)$$

where $\Delta\theta_{-j}^{[t]} = \theta_{-j}^{[t]} - \theta^{[t]}$.

Substituting $q(t) = \nabla_x \nabla_\theta \ell(z_{\text{test}}; \theta^{[t]})$:

$$Q_{-j}^{[t_1,t_2]}(q) = \left\langle \nabla_\theta \nabla_x \ell(z_{\text{test}}; \theta^{[t_2]}), \theta_{-j}^{[t_2]} - \theta^{[t_2]} \right\rangle - \left\langle \nabla_\theta \nabla_x \ell(z_{\text{test}}; \theta^{[t_1]}), \theta_{-j}^{[t_1]} - \theta^{[t_1]} \right\rangle. \tag{54}$$

We apply the first-order Taylor approximation of $\nabla_x \ell(z_{\text{test}}; \theta)$ around $\theta^{[t_2]}$ and $\theta^{[t_1]}$:

$$\nabla_x \ell(z_{\text{test}}; \theta_{-j}^{[t_2]}) \approx \nabla_x \ell(z_{\text{test}}; \theta^{[t_2]}) + \nabla_\theta \nabla_x \ell(z_{\text{test}}; \theta^{[t_2]}) \left( \theta_{-j}^{[t_2]} - \theta^{[t_2]} \right), \tag{55}$$

$$\nabla_x \ell(z_{\text{test}}; \theta_{-j}^{[t_1]}) \approx \nabla_x \ell(z_{\text{test}}; \theta^{[t_1]}) + \nabla_\theta \nabla_x \ell(z_{\text{test}}; \theta^{[t_1]}) \left( \theta_{-j}^{[t_1]} - \theta^{[t_1]} \right). \tag{56}$$

Rearranging these equations:

$$\left\langle \nabla_\theta \nabla_x \ell(z_{\text{test}}; \theta^{[t_2]}), \theta_{-j}^{[t_2]} - \theta^{[t_2]} \right\rangle \approx \nabla_x \ell(z_{\text{test}}; \theta_{-j}^{[t_2]}) - \nabla_x \ell(z_{\text{test}}; \theta^{[t_2]}), \tag{57}$$

$$\left\langle \nabla_\theta \nabla_x \ell(z_{\text{test}}; \theta^{[t_1]}), \theta_{-j}^{[t_1]} - \theta^{[t_1]} \right\rangle \approx \nabla_x \ell(z_{\text{test}}; \theta_{-j}^{[t_1]}) - \nabla_x \ell(z_{\text{test}}; \theta^{[t_1]}). \tag{58}$$

Substituting these approximations back into Eq.(54):

$$Q_{-j}^{[t_1,t_2]}(q) \approx \left[ \nabla_x \ell(z_{\text{test}}; \theta_{-j}^{[t_2]}) - \nabla_x \ell(z_{\text{test}}; \theta^{[t_2]}) \right] - \left[ \nabla_x \ell(z_{\text{test}}; \theta_{-j}^{[t_1]}) - \nabla_x \ell(z_{\text{test}}; \theta^{[t_1]}) \right]$$

$$= \left[ \nabla_x \ell(z_{\text{test}}; \theta_{-j}^{[t_2]}) - \nabla_x \ell(z_{\text{test}}; \theta_{-j}^{[t_1]}) \right] - \left[ \nabla_x \ell(z_{\text{test}}; \theta^{[t_2]}) - \nabla_x \ell(z_{\text{test}}; \theta^{[t_1]}) \right]. \tag{59}$$

This completes the proof. $\qquad\square$

This theorem shows how DIT measures the impact of excluding a training sample $z_j$ on the gradient of the loss with respect to the input features at the test point $z_{\text{test}}$ over the interval $[t_1, t_2]$. This provides insights into how the importance of different input features evolves during training and how individual training samples influence this feature importance.

### A.3.4 DIT FOR PARAMETER IMPORTANCE

**Theorem 10** (DIT for Parameter Importance). Given a model with parameters $\theta \in \mathbb{R}^p$, a time window $[t_1, t_2]$, a training sample $z_j$, and the $i$-th standard basis vector $e_i \in \mathbb{R}^p$, the Dynamic Influence Tracker with query function $q(t) = (e_i)$ is exactly:

$$Q_{-j}^{[t_1,t_2]}(q) = \left( \theta_{-j,i}^{[t_2]} - \theta_{-j,i}^{[t_1]} \right) - \left( \theta_i^{[t_2]} - \theta_i^{[t_1]} \right), \tag{60}$$

where $\theta_{-j,i}^{[t]}$ denotes the $i$-th component of the model parameters at time $t$ when trained without sample $z_j$, and $\theta_i^{[t]}$ denotes the $i$-th component of the parameters when trained with all samples.

*Proof.* We start with the definition of the Query-Based Dynamic Influence Tracker:

$$Q_{-j}^{[t_1,t_2]}(q) = \left\langle q(t_2), \Delta\theta_{-j}^{[t_2]} \right\rangle - \left\langle q(t_1), \Delta\theta_{-j}^{[t_1]} \right\rangle, \tag{61}$$

where $\Delta\theta_{-j}^{[t]} = \theta_{-j}^{[t]} - \theta^{[t]}$.

Substituting $q(t) = e_i$, which is constant over time:

$$Q_{-j}^{[t_1,t_2]}(q) = \left\langle e_i, \theta_{-j}^{[t_2]} - \theta^{[t_2]} \right\rangle - \left\langle e_i, \theta_{-j}^{[t_1]} - \theta^{[t_1]} \right\rangle. \tag{62}$$

Since $e_i$ is the $i$-th standard basis vector, the inner product selects the $i$-th component:

$$Q_{-j}^{[t_1,t_2]}(q) = \left( \theta_{-j,i}^{[t_2]} - \theta_i^{[t_2]} \right) - \left( \theta_{-j,i}^{[t_1]} - \theta_i^{[t_1]} \right) = \left( \theta_{-j,i}^{[t_2]} - \theta_{-j,i}^{[t_1]} \right) - \left( \theta_i^{[t_2]} - \theta_i^{[t_1]} \right). \tag{63}$$

This matches the expression in Eq. (60), completing our proof. $\qquad\square$

This theorem allows us to isolate the influence of a training sample $z_j$ on specific model parameters over the interval $[t_1, t_2]$. A large absolute value of $Q_{-j}^{[t_1, t_2]}(q)$ indicates that sample $z_j$ has a significant influence on the $i$-th parameter during the specified time window. This is particularly useful for identifying which parameters are most affected by specific training samples and understanding the localized effects of training samples on the model.

By analyzing how $Q_{-j}^{[t_1, t_2]}(q)$ changes over different time windows, we can understand how the influence of a training sample on specific parameters evolves throughout the training process.

## A.4 Estimation Error Analysis without Convexity Assumptions

**Theorem 11** (Error Bound for DIT Parameter Change). Let $\Delta\theta_{-j}^{[t_1,t_2]}$ be the true influence of excluding sample $z_j$ on the model parameters over the interval $[t_1, t_2]$ during SGD training. Let $\widehat{\Delta\theta}_{-j}^{[t_1,t_2]}$ be its approximation using DIT. Under the following assumptions:

(A1) Lipschitz Continuity of Gradient: The gradient $\nabla\ell(z_i; \theta)$ is Lipschitz continuous with constant $L_g$: $\|\nabla\ell(z_i; \theta_1) - \nabla\ell(z_i; \theta_2)\| \leq L_g \|\theta_1 - \theta_2\|$, $\forall\theta_1, \theta_2 \in \Theta, \forall i$.

(A2) Lipschitz Continuity of Hessian: The Hessian $\nabla^2\ell(z_i; \theta)$ is Lipschitz continuous with constant $L_H$: $\|\nabla^2\ell(z_i; \theta_1) - \nabla^2\ell(z_i; \theta_2)\| \leq L_H \|\theta_1 - \theta_2\|$, $\forall\theta_1, \theta_2 \in \Theta, \forall i$.

(A3) Learning Rate Bound: The learning rate satisfies $\eta_t \leq \frac{1}{L_H}$ for all $t$.

(A4) Hessian Approximation Error: The Hessian approximation error is bounded: $\|H^{[t]} - H_{-j}^{[t]}\| \leq \epsilon_H$, $\forall t$, where $H_{-j}^{[t]} = \frac{1}{|S_t \setminus \{j\}|}\sum_{i \in S_t \setminus \{j\}} \nabla^2\ell(z_i; \theta^{[t]})$ is the empirical Hessian over the mini-batch.

(A5) Gradient Norm Bound: For all $\theta \in \Theta$ and all $z_i$: $\|\nabla\ell(z_i; \theta)\| \leq G$.

(A6) Parameter Difference Bound: There exists a constant $M > 0$ such that: $\|\theta_{-j}^{[t]} - \theta^{[t]}\| \leq M$, $\forall t \in [t_1, t_2]$.

(A7) Bounded Hessian Norm: For all $\theta \in \Theta$ and all $z_i$: $\|\nabla^2\ell(z_i; \theta)\| \leq M_H$.

Then, the expected estimation error is bounded as follows:

$$\mathbb{E}\left[\left\|\Delta\theta_{-j}^{[t_1,t_2]} - \widehat{\Delta\theta}_{-j}^{[t_1,t_2]}\right\|\right] \leq \frac{\tilde{B}}{M_H}\left(e^{M_H\eta_{\max}(t_2+1)} + e^{M_H\eta_{\max}(t_1+1)} - 2\right) \tag{64}$$

where: $\eta_{\max} = \max_{t \in [t_1,t_2]} \eta_t$, $\tilde{B} = \frac{L_H M^2}{2} + \epsilon_H M$, $n$ is the total number of samples in the dataset.

*Proof.* **Step 1: Derivation of the Error Update Equation**

Define the error at iteration $t$:

$$e^{[t]} = (\theta_{-j}^{[t]} - \theta^{[t]}) - \widehat{\Delta\theta}_{-j}^{[t]} \tag{65}$$

where $\widehat{\Delta\theta}_{-j}^{[t]} = \widehat{\Delta\theta}_{-j}^{[0,t]}$ is the approximation of the true parameter change $\Delta\theta_{-j}^{[t]}$ using the DIT method.

Our aim is to derive a recursive relation for $e^{[t]}$ and then bound its expected norm.

Consider the updates for $\theta^{[t]}$, $\theta_{-j}^{[t]}$, and $\widehat{\theta}_{-j}^{[t]}$:

Original SGD Update:

$$\theta^{[t+1]} = \theta^{[t]} - \eta_t \tilde{g}^{[t]}, \quad \tilde{g}^{[t]} = \frac{1}{|S_t|}\sum_{i \in S_t} \nabla\ell(z_i; \theta^{[t]}). \tag{66}$$

Leave-One-Out SGD Update:

$$\theta_{-j}^{[t+1]} = \theta_{-j}^{[t]} - \eta_t \tilde{g}_{-j}^{[t]}, \quad \tilde{g}_{-j}^{[t]} = \frac{1}{|S_t|}\sum_{i \in S_t \setminus \{j\}} \nabla\ell(z_i; \theta_{-j}^{[t]}). \tag{67}$$

Approximate Leave-One-Out Update (DIT Method):

$$\widehat{\theta}_{-j}^{[t+1]} = \widehat{\theta}_{-j}^{[t]} - \eta_t \left(\tilde{g}^{[t]} + H^{[t]}(\widehat{\theta}_{-j}^{[t]} - \theta^{[t]}) - \mathbf{1}_{\{j \in S_t\}}\frac{1}{|S_t|}\nabla\ell(z_j; \theta^{[t]})\right). \tag{68}$$

We derive the error update equation as follows:

$$e^{[t]} - e^{[t-1]} = \eta_{t-1}\delta^{[t-1]}, \tag{69}$$

where:

$$\delta^{[t-1]} = \left(\tilde{g}_{-j}^{[t-1]} - \tilde{g}^{[t-1]}\right) - H^{[t-1]}\widehat{\Delta\theta}_{-j}^{[t-1]} + \mathbf{1}_{\{j \in S_{t-1}\}}\frac{1}{|S_{t-1}|}\nabla\ell(z_j;\theta^{[t-1]}). \tag{70}$$

or equivalently:

$$\delta^{[t]} = \left(\tilde{g}_{-j}^{[t]} - \tilde{g}^{[t]}\right) - H^{[t]}\widehat{\Delta\theta}_{-j}^{[t]} + \mathbf{1}_{\{j \in S_t\}}\frac{1}{|S_t|}\nabla\ell(z_j;\theta^{[t]}). \tag{71}$$

**Step 2: Bounding $\|\delta^{[t]}\|$**

We decompose $\delta^{[t]}$ and bound each term:

**1. Difference in Stochastic Gradients:**

$$\tilde{g}_{-j}^{[t]} - \tilde{g}^{[t]} = \frac{1}{|S_t|}\left(\sum_{i \in S_t \setminus \{j\}}\left(\nabla\ell(z_i;\theta_{-j}^{[t]}) - \nabla\ell(z_i;\theta^{[t]})\right) - \mathbf{1}_{\{j \in S_t\}}\nabla\ell(z_j;\theta^{[t]})\right). \tag{72}$$

Applying a first-order Taylor expansion to $\nabla\ell(z_i;\theta_{-j}^{[t]})$ for $i \neq j$:

$$\nabla\ell(z_i;\theta_{-j}^{[t]}) - \nabla\ell(z_i;\theta^{[t]}) = \nabla^2\ell(z_i;\theta^{[t]})(\theta_{-j}^{[t]} - \theta^{[t]}) + r_{i,j}^{[t]}, \tag{73}$$

where, by Assumption (A2):

$$\|r_{i,j}^{[t]}\| \leq \frac{L_H}{2}\|\theta_{-j}^{[t]} - \theta^{[t]}\|^2 \tag{74}$$

Thus, we have:

$$\tilde{g}_{-j}^{[t]} - \tilde{g}^{[t]} = \frac{1}{|S_t|}\sum_{i \in S_t \setminus \{j\}}\nabla^2\ell(z_i;\theta^{[t]})(\theta_{-j}^{[t]} - \theta^{[t]}) + r_{i,j}^{[t]} - \mathbf{1}_{\{j \in S_t\}}\nabla\ell(z_j;\theta^{[t]})$$

$$= \frac{1}{|S_t|}\left(\sum_{i \in S_t \setminus \{j\}}r_{i,j}^{[t]} - \mathbf{1}_{\{j \in S_t\}}\nabla\ell(z_j;\theta^{[t]})\right) + H_{-j}^{[t]}(\theta_{-j}^{[t]} - \theta^{[t]}) \tag{75}$$

**2. Hessian Approximation Error:**

$$\|(H_{-j}^{[t]} - H^{[t]})(\theta_{-j}^{[t]} - \theta^{[t]})\| \leq \epsilon_H\|\theta_{-j}^{[t]} - \theta^{[t]}\|. \tag{76}$$

according to Assumption (A4).

**3. Combining Terms:** Substitute the approximations back into $\delta^{[t]}$:

$$\delta^{[t]} = \left(\tilde{g}_{-j}^{[t]} - \tilde{g}^{[t]}\right) - H^{[t]}\widehat{\Delta\theta}_{-j}^{[t]} + \mathbf{1}_{\{j \in S_t\}}\frac{1}{|S_t|}\nabla\ell(z_j;\theta^{[t]})$$

$$= \left(\tilde{g}_{-j}^{[t]} - \tilde{g}^{[t]}\right) - H_{-j}^{[t]}(\theta_{-j}^{[t]} - \theta^{[t]}) + \left(H_{-j}^{[t]} - H^{[t]}\right)(\theta_{-j}^{[t]} - \theta^{[t]}) + \mathbf{1}_{\{j \in S_t\}}\frac{1}{|S_t|}\nabla\ell(z_j;\theta^{[t]})$$

$$= \frac{1}{|S_t|}\sum_{i \in S_t \setminus \{j\}}r_{i,j}^{[t]} + (H_{-j}^{[t]} - H^{[t]})(\theta_{-j}^{[t]} - \theta^{[t]}) + H^{[t]}((\theta_{-j}^{[t]} - \theta^{[t]}) - \Delta\widehat{\theta}_{-j}^{[t]})$$

$$= \frac{1}{|S_t|}\sum_{i \in S_t \setminus \{j\}}r_{i,j}^{[t]} + (H_{-j}^{[t]} - H^{[t]})(\theta_{-j}^{[t]} - \theta^{[t]}) + H^{[t]}e^{[t]}. \tag{77}$$

**4. Bounding $\|\delta^{[t]}\|$:**

- **First Term:**

$$\left\|\frac{1}{|S_t|}\sum_{i \in S_t \setminus \{j\}}r_{i,j}^{[t]}\right\| < \frac{L_H M^2}{2}. \tag{78}$$

- **Second Term:**

$$\left\| (H_{-j}^{[t]} - H^{[t]})(\theta_{-j}^{[t]} - \theta^{[t]}) \right\| \le \epsilon_H M. \tag{79}$$

- **Third Term:**

$$\left\| H^{[t]} e^{[t]} \right\| \le M_H \| e^{[t]} \|. \tag{80}$$

Combining bounds, we can have:

$$\| \delta^{[t]} \| < \frac{L_H M^2}{2} + \epsilon_H M + M_H \| e^{[t]} \|. \tag{81}$$

**Step 3: Error Update Equation**

Using the error update:

$$e^{[t]} = e^{[t-1]} - \eta_t \delta^{[t-1]}, \tag{82}$$

we have:

$$\| e^{[t]} \| \le \| e^{[t-1]} \| + \eta_t \| \delta^{[t-1]} \| < \| e^{[t-1]} \| + \eta_t \left( \frac{L_H M^2}{2} + \epsilon_H M + M_H \| e^{[t-1]} \| \right). \tag{83}$$

Define:

$$a_t = 1 + \eta_t M_H, \quad b_t = \eta_t \left( \frac{L_H M^2}{2} + \epsilon_H M \right). \tag{84}$$

Then:

$$\| e^{[t]} \| < a_t \| e^{[t-1]} \| + b_t. \tag{85}$$

**Step 4: Taking Expectations**

Taking expectations over the mini-batch sampling:

$$\mathbb{E}\left[ \| e^{[t]} \| \right] < a_t \mathbb{E}\left[ \| e^{[t-1]} \| \right] + b_t. \tag{86}$$

Define:

$$\tilde{B} = \frac{L_H M^2}{2} + \epsilon_H M. \tag{87}$$

Then:

$$\mathbb{E}\left[ \| e^{[t]} \| \right] < a_t \mathbb{E}\left[ \| e^{[t-1]} \| \right] + \eta_t \tilde{B}. \tag{88}$$

**Step 5: Solving the Recurrence Relation**

Unfolding the recurrence:

$$\mathbb{E}\left[ \| e^{[t]} \| \right] \le \prod_{k=0}^{t} a_k \cdot \mathbb{E}\left[ \| e^{[0]} \| \right] + \sum_{s=0}^{t} \left( \prod_{k=s+1}^{t} a_k \right) b_s. \tag{89}$$

Since $e^{[0]} = 0$, we have:

$$\mathbb{E}\left[ \| e^{[t]} \| \right] \le \sum_{s=0}^{t} \left( \prod_{k=s+1}^{t} a_k \right) b_s. \tag{90}$$

Assuming $a_k \le e^{M_H \eta_{\max}}$, we get:

$$\prod_{k=s+1}^{t} a_k \le e^{M_H \eta_{\max}(t-s)}. \tag{91}$$

Therefore:

$$\mathbb{E}\left[ \| e^{[t]} \| \right] \le \tilde{B} \eta_{\max} \sum_{s=0}^{t} e^{M_H \eta_{\max}(t-s)}. \tag{92}$$

Approximating the sum:

$$\mathbb{E}\left[\|e^{[t]}\|\right] \le \tilde{B}\eta_{\max} \cdot \frac{e^{M_H\eta_{\max}(t+1)} - 1}{e^{M_H\eta_{\max}} - 1}. \tag{93}$$

For small $M_H\eta_{\max}$, $e^{M_H\eta_{\max}} - 1 \approx M_H\eta_{\max}$, yielding:

$$\mathbb{E}\left[\|e^{[t]}\|\right] \le \frac{\tilde{B}}{M_H}\left(e^{M_H\eta_{\max}(t+1)} - 1\right). \tag{94}$$

Substitute $t$ with $t_1$ and $t_2$ respectively:

$$\mathbb{E}\left[\|e^{[t_2]}\|\right] \le \frac{\tilde{B}}{M_H}\left(e^{M_H\eta_{\max}(t_2+1)} - 1\right), \tag{95}$$

$$\mathbb{E}\left[\|e^{[t_1]}\|\right] \le \frac{\tilde{B}}{M_H}\left(e^{M_H\eta_{\max}(t_1+1)} - 1\right). \tag{96}$$

**Step 6: Final Bound**

The estimation error is:

$$\mathbb{E}\left[\left\|\Delta\theta_{-j}^{[t_1,t_2]} - \widehat{\Delta\theta}_{-j}^{[t_1,t_2]}\right\|\right] \le \mathbb{E}\left[\|e^{[t_2]}\|\right] + \mathbb{E}\left[\|e^{[t_1]}\|\right]$$

$$\le \frac{\tilde{B}}{M_H}\left(e^{M_H\eta_{\max}(t_2+1)} + e^{M_H\eta_{\max}(t_1+1)} - 2\right) \tag{97}$$

This completes the proof. □

**Remark 12.** The error bound provides several key insights:

- The error grows at most exponentially with both $t_1$ and $t_2$, highlighting the challenge of long-range influence estimation. The impact of $t_2$ is generally more significant as it represents the end of the time window.

- The Hessian approximation error $\epsilon_H$ directly impacts the overall error, emphasizing the importance of accurate Hessian estimation.

- The maximum learning rate $\eta_{\max}$ affects the error bound exponentially, suggesting that smaller learning rates might help control the estimation error.

- The bound depends on the Lipschitz constants of the gradient and Hessian ($L_g$ and $L_H$), indicating that smoother loss landscapes lead to more reliable influence estimates.

This theorem provides a theoretical foundation for understanding the limitations of influence estimation without assuming convexity and guides practical considerations in its application to large-scale machine learning problems.

## A.5 PROOF OF ALGORITHM 2

We begin by recalling the definition:

$$Q_{-j}^{[t_1,t_2]}(q) = \langle q(t_2), \Delta\theta_{-j}^{[t_2]} \rangle - \langle q(t_1), \Delta\theta_{-j}^{[t_1]} \rangle \tag{98}$$

where $\Delta\theta_{-j}^{[t]} \approx \sum_{s=0}^{t-1} \left( \prod_{k=s+1}^{t-1} Z_k \right) \tilde{\mathbf{1}}_j^{[s]}$, and $Z_t = I - \eta_t H^{[t]}$, $\tilde{\mathbf{1}}_j^{[t]} = \mathbf{1}_{j \in S_t} \frac{\eta_t}{|S_t|} g(z_j; \theta^{[t]})$.

Note that $Z_t$ is self-adjoint matrix, adhering to $\langle x, Z_t y \rangle = \langle Z_t x, y \rangle$ for all vectors $x, y$.

According to the update rules for $u_1$ and $u_2$ in the algorithm:

$$u_i^{[t-1]} = u_i^{[t]} - \eta_t H^{[t]} u_i^{[t]} = (I - \eta_t H^{[t]}) u_i^{[t]} = Z_t u_i^{[t]}, \quad i \in \{1, 2\} \tag{99}$$

By recursive application of this update rule, we obtain for $s < t$:

$$u_i^{[s]} = \left( \prod_{k=s+1}^{t-1} Z_k \right) u_i^{[t]}, \quad i \in \{1, 2\} \tag{100}$$

According to the accumulation of $Q$ in the algorithm, at each time step $t$, if $j \in S_t$, we have:

$$\Delta Q_t = \left\langle (u_2^{[t]} - u_1^{[t]}), \frac{\eta_t}{|S_t|} g(z_j; \theta^{[t]}) \right\rangle \tag{101}$$

The algorithm initializes $u_2^{[t_2-1]} = q(t_2)$ and sets $u_1^{[t_1-1]} = q(t_1)$ at time $t_1$. Importantly, $u_1$ is not updated beyond $t_1$. Using the result from Eq. (100), we can express $u_2^{[t]}$ and $u_1^{[t]}$ as:

$$u_2^{[t]} = \prod_{k=t+1}^{t_2-1} Z_k q(t_2), \quad \text{for } 0 \le t < t_2 \tag{102}$$

$$u_1^{[t]} = \begin{cases} \prod_{k=t+1}^{t_1-1} Z_k q(t_1) & \text{for } 0 \le t < t_1 \\ 0 & \text{for } t_1 \le t < t_2 \end{cases} \tag{103}$$

Note that $u_1^{[t]} = 0$ for $t_1 \le t < t_2$ because $u_1$ is not updated beyond $t_1$, effectively removing its contribution to $\Delta Q_t$ in this range.

Substituting these expressions into Eq.(101):

$$\Delta Q_t = \begin{cases} \left\langle \prod_{k=t+1}^{t_2-1} Z_k q(t_2) - \left( \prod_{k=t+1}^{t_1-1} Z_k q(t_1) \right), \tilde{\mathbf{1}}_j^{[t]} \right\rangle & \text{for } 0 \le t < t_1 \\ \left\langle \prod_{k=t+1}^{t_2-1} Z_k q(t_2), \tilde{\mathbf{1}}_j^{[t]} \right\rangle & \text{for } t_1 \le t < t_2 \end{cases} \tag{104}$$

The total $Q$ is the sum of all $\Delta Q_t$: $Q = \sum_{t=0}^{t_2-1} \Delta Q_t$.

Expanding this sum and recalling that $Z_t$ is self-adjoint, we get:

$$Q = \left\langle q(t_2), \sum_{t=0}^{t_2-1} \left( \prod_{k=t+1}^{t_2-1} Z_k \right) \tilde{\mathbf{1}}_j^{[t]} \right\rangle - \left\langle q(t_1), \sum_{t=0}^{t_1-1} \left( \prod_{k=t+1}^{t_1-1} Z_k \right) \tilde{\mathbf{1}}_j^{[t]} \right\rangle \tag{105}$$

Note that $u_2^{[t]}$ contributes to the first term over the entire interval $[0, t_2)$, while $u_1^{[t]}$ only contributes to the second term over $[0, t_1)$. This distinction arises from the algorithm's design, where $u_1$ is not updated beyond $t_1$.

Combined Eq. (105) are precisely the definitions of $\Delta\theta_{-j}^{[t_2]}$ and $\Delta\theta_{-j}^{[t_1]}$, we have:

$$Q = \langle q(t_2), \Delta\theta_{-j}^{[t_2]} \rangle - \langle q(t_1), \Delta\theta_{-j}^{[t_1]} \rangle = Q_{-j}^{[t_1,t_2]}(q) \tag{106}$$

Thus, we have rigorously demonstrated that the algorithm's output $Q$ is equivalent to the defined $Q_{-j}^{[t_1,t_2]}(q)$ in Eq. (98) under the stated assumption on $\eta_t$.

A.6 EXPERIMENTAL SUPPLEMENT

A.6.1 EXPERIMENTAL SETUP

**Datasets** We employed four diverse datasets spanning various domains and complexities to evaluate the robustness and generalizability of DIT.

- **Adult** (Dua & Graff, 2019): A dataset for income prediction containing 48,842 instances with 14 mixed categorical and numerical features. The dataset is preprocessed by handling missing values, normalizing numerical features, and applying one-hot encoding to categorical features. The task is a binary classification of predicting whether income exceeds $50K/year.

- **20 Newsgroups** (Lang, 1995): A text classification dataset. Text data is converted to TF-IDF vectors, and stop words are removed for cleaner feature representation. We focus on binary classification between categories *comp.sys.ibm.pc.hardware* and *comp.sys.mac.hardware*. The task is to classify posts into one of the two hardware categories.

- **MNIST** (LeCun et al., 2010): A dataset of 70,000 handwritten digit images, each 28x28 pixels in grayscale. Binary classification is conducted between digits *1* and *7*, where the pixel intensities are normalized.

- **EMNIST** (Cohen et al., 2017): An extended MNIST dataset for handwritten letters. The grayscale images are normalized to ensure uniformity in the input space. We focus on binary classification between letters *A* and *B*.

**Model Architectures** We implemented three model architectures of varying complexity to evaluate the performance of DIT across different learning paradigms. In all models, the final layer outputs a single value for binary classification, and all use binary cross-entropy loss with logits.

- **Logistic Regression (LR)**: Implemented as a single-layer neural network without hidden layers. The input dimension is flattened to accommodate various input shapes.

- **Deep Neural Network (DNN)**: The architecture comprises two hidden layers, each with eight units followed by a ReLU activation function. The second layer outputs a single value for binary classification. The input is flattened, similar to logistic regression.

- **Convolutional Neural Network (CNN)**: This architecture is used for image datasets like MNIST and EMNIST. It consists of two convolutional layers, with 32 and 64 filters, respectively, each followed by ReLU activation and max-pooling. The final output from the convolutional layers is flattened and passed through a linear layer to output a binary classification value.

For non-image data like Adult and 20 Newsgroups, the input is a vector, while image data like MNIST and EMNIST is reshaped into a single dimension for LR and DNN models. The CNN processes image data in its original 2D format.

**Evaluation Metrics** To comprehensively evaluate the performance of DIT, we employed a suite of statistical metrics, each capturing different aspects of the relationship between compared methods:

- **Pearson Correlation Coefficient ($r$)** (Pearson, 1895): The Pearson correlation coefficient measures the linear relationship between two variables. For two sets of data, X and Y, it is calculated as:

$$r = \frac{\sum_{i=1}^{n}(X_i - \bar{X})(Y_i - \bar{Y})}{\sqrt{\sum_{i=1}^{n}(X_i - \bar{X})^2 \sum_{i=1}^{n}(Y_i - \bar{Y})^2}}$$

where $\bar{X}$ and $\bar{Y}$ are the means of X and Y respectively, and $n$ is the number of samples. This metric is valuable for identifying direct proportional or inversely proportional relationships within the data. $r$ ranges from -1 to 1, where 1 indicates a perfect positive linear relationship, -1 indicates a perfect negative linear relationship, and 0 indicates no linear relationship.

- **Spearman's Rank Correlation Coefficient** ($\rho$) (Spearman, 1987): Spearman's rank correlation assesses monotonic relationships by comparing the rank orders of samples:

$$\rho = 1 - \frac{6 \sum_{i=1}^{n} d_i^2}{n(n^2 - 1)}$$

where $d_i$ is the difference between the ranks of corresponding values $X_i$ and $Y_i$, and $n$ is the number of samples. $\rho$ ranges from -1 to 1, with values close to 1 or -1 indicating strong monotonic relationships (positive or negative, respectively) and values close to 0 indicating weak monotonic relationships.

- **Kendall's Tau** ($\tau$) (Kendall, 1938): Kendall's Tau evaluates ordinal relationships by measuring the number of concordant and discordant pairs:

$$\tau = \frac{2(n_c - n_d)}{n(n - 1)}$$

where $n_c$ is the number of concordant pairs, $n_d$ is the number of discordant pairs, and $n$ is the total number of pairs. $\tau$ ranges from -1 to 1, with 1 indicating perfect agreement between two rankings, -1 indicating perfect disagreement, and 0 indicating no relationship.

- **Jaccard Similarity** ($J$) (Jaccard, 1912): The Jaccard similarity coefficient compares the overlap between the top 30% of influential points as determined by different methods:

$$J(A, B) = \frac{|A \cap B|}{|A \cup B|}$$

where $A$ and $B$ are the sets of top 30% influential points identified by different methods. $J$ ranges from 0 to 1, with 1 indicating perfect overlap between the sets and 0 indicating no overlap.

By capturing linear relationships (Pearson), monotonic relationships (Spearman), ordinal relationships (Kendall's Tau), and set-based similarities (Jaccard), we ensure a multifaceted evaluation of influence estimation methods.

To ensure transparency and reproducibility, all code, including detailed hyperparameter settings and training procedures, is available on our GitHub repository. This repository contains scripts and configuration files that define the exact setup for each model used in our experiments, encompassing learning rates, batch sizes, regularization strategies, and any other model-specific training details.

### A.6.2 SAMPLE INFLUENCE DYNAMICS METHODOLOGY

The methodology for analyzing sample influence dynamics consists of several key steps.

1) **Sampling and Influence Tracking**: We randomly select 256 training points and track their influence, measured as loss change via LOO, over 20 epochs of training. This fine-grained sampling provides detailed influence trajectories for each point.

2) **Standardization and Trend Analysis**: We standardize the influence values using StandardScaler to normalize the value across different epochs. For each sample, a linear regression is performed on its standardized influence values over time. The slope of this regression line indicates the overall trend direction (increasing or decreasing influence). The p-value of the regression determines whether this trend is statistically significant.

3) **Adaptive Pattern Categorization**: Each sample is categorized based on its statistical properties, including a) Trend significance (determined by the p-value) b) Trend direction (positive or negative slope) c) Standard deviation of influence values (a measure of fluctuation).

4) **Pattern Analysis**: We calculate the proportion of samples in each category and compute the centroid of each category by averaging the standardized influence values of all points within that category.

### A.6.3 IDENTIFICATION OF TRAINING STAGES

To identify stages in the training process, we utilized the following method:

1) **Modeling Loss Trajectory**: We analyzed the loss trajectory across epochs by fitting an exponential decay model. This approach helps to smooth out fluctuations and emphasize underlying trends in the training loss.

2) **Residual Calculation**: Residuals were computed as the differences between the actual loss values and the values predicted by the exponential model. These residuals highlight where the actual training deviates from the predicted trend.

3) **Change Point Detection**: We identified peaks in the absolute residuals as change points. A minimum distance criterion was applied to ensure these change points were evenly distributed across the training timeline.

4) **Stage Segmentation**: Based on the identified change points, the training process was divided into three stages: early, middle, and late.

