# OpenReview forum: "Dynamic Influence Tracker: Estimating Sample Influence in SGD-Trained Models across Arbitrary Time Windows"
_ICLR.cc/2025/Conference — ICLR 2025 Conference Withdrawn Submission_

### Official Review · Reviewer_5BaV · 2024-11-03

**Soundness:** 1
**Presentation:** 3
**Contribution:** 2
**Rating:** 3
**Confidence:** 4

**Summary:**

This paper proposes a method, called Dynamic Influence Tracker (DIT), to estimate sample influence during the training process of SGD. The authors provide an error bound for their estimator and show DIT's effectiveness in various examples including anomalous sample detection and curriculum learning.

**Strengths:**

The authors proposed a method to track the sample influence during a training process. This approach allows us to discover samples that are important in the early stage of training but become dispensable in the end, whereas existing approaches focus on assessing the sample influence for the optimizer. They justified their approach from both theoretical and empirical perspectives.

**Weaknesses:**

I have two major concerns regarding this paper.

1. In the last term of Eq. (14) and subsequent definition of $\tilde{\mathbf{1}_j^{[t]}}$, the $\theta^{[t]}$ inside the function $g$ should be $\theta_{-j}^{[t]}$. This is not just a typo since it also exists in the proofs (e.g., the last term of Eq. (57)) and Alg. 2. Hence, the whole method requires significant revision.

2. The paper didn't discuss the literature properly which put their claimed contributions in question.
    1. On the computational aspect, the main criticism the authors hold on the influence function is the computation of inverse Hessian, and they propose to project the influence function onto a query function. However, on the one hand, there have been methods designed to reduce this computational complexity such as Arnoldi iteration [1]; on the other hand, the idea of projection has already appeared in the Maximum Influence Subset framework [2].
    2. The definition of influence function does not require convexity. Convexity is only assumed when analyzing its statistical properties (see [3] for an example). Even though convexity is not assumed in the proofs, boundedness of gradient and hessian are assumed instead, which are very strong. In [3], only tail assumptions are made, so it is unclear whether the authors' argument requires less stringent assumptions.

[1] A. Schioppa, et. al. Scaling up influence functions. AAAI, 2022.
[2] T. Broderick, R. Giordano, and R. Meager. An Automatic Finite-Sample Robustness Metric: When Can Dropping a Little Data Make a Big Difference. Arxiv, 2020.
[3] J. Fisher et. al. Influence Diagnostics under Self-concordance. AISTATS, 2023.

**Questions:**

See the comments in the previous section.

---

> ### Author Response · Authors · 2024-11-13
> **Step-by-step Derivation Proof.**
>
> ## Concern 1:
>
> We sincerely appreciate the careful review. Below is a **step-by-step derivation to address your concerns.**
> We start from Eq.(10), which establishes the relationship (denoted as Eq.(10-1)):
> $$
> \theta_{-j}^{[t+1]} - \theta^{[t+1]} = (\theta_{-j}^{[t]} - \theta^{[t]}) - \frac{\eta_t}{|S_t|} (\sum_{i \in S_t\setminus\{j\}} g(z_i; \theta_{-j}^{[t]}) - \sum_{i \in S_t} g(z_i; \theta^{[t]}))
> $$
>
> $$
> = (\theta_{-j}^{[t]} - \theta^{[t]}) - \frac{\eta_t}{|S_t|} (\sum_{i \in S_t\setminus\{j\}} g(z_i; \theta_{-j}^{[t]}) -\sum_{i \in S_t\setminus\{j\}}g(z_i; \theta^{[t]})-\mathbf{1}_{j \in S_t}g(z_i; \theta^{[t]}))
> $$
>
> $$
> = (\theta_{-j}^{[t]} - \theta^{[t]}) - \frac{\eta_t}{|S_t|} (\sum_{i \in S_t\setminus\{j\}} g(z_i; \theta_{-j}^{[t]}) -\sum_{i \in S_t\setminus\{j\}}g(z_i; \theta^{[t]}))+\frac{\eta_t}{|S_t|}\mathbf{1}_{j \in S_t}g(z_i; \theta^{[t]})
> $$
>
> $$
> = (\theta_{-j}^{[t]} - \theta^{[t]}) - \frac{\eta_t}{|S_t|} \sum_{i \in S_t\setminus\{j\}} (g(z_i; \theta_{-j}^{[t]}) -g(z_i; \theta^{[t]}))+\frac{\eta_t}{|S_t|}\mathbf{1}_{j \in S_t}g(z_i; \theta^{[t]}),
> $$
>
> where $\mathbf{1}_{j \in S_t}$ is an indicator function that equals 1 if $j \in S_t$, otherwise 0.
>
> Using Eq. (11), we have (denoted as Eq.(11-1)):
>
> $$
> \sum_{i \in S_t\setminus\{j\}}( g(z_i; \theta_{-j}^{[t]}) - g(z_i; \theta^{[t]}) )\approx \sum_{i \in S_t\setminus\{j\}} \nabla_\theta g(z_i; \theta^{[t]})^T (\theta_{-j}^{[t]} - \theta^{[t]}),
> $$
>
> Following Eq. (12) and Assumption (A4), we have (denoted as Eq.(12-1)):
>
> $$
> \sum_{i\in S_t\backslash\{j\}} \nabla_\theta g(z_i; \theta^{[t]})^T(\theta_{-j}^{[t]} - \theta^{[t]})=|S_t| H_{-j}^{[t]}(\theta_{-j}^{[t]} - \theta^{[t]})\approx |S_t|  H^{[t]}(\theta_{-j}^{[t]} - \theta^{[t]}).
> $$
>
> Combining Eq.(11-1) and Eq.(12-1), we have (denoted as Eq.(11-2)):
>
> $$
> \sum_{i \in S_t\setminus\{j\}}( g(z_i; \theta_{-j}^{[t]}) - g(z_i; \theta^{[t]}) )\approx |S_t|  H^{[t]}(\theta_{-j}^{[t]} - \theta^{[t]}).
> $$
>
> Applying Eq.(11-2) to Eq.(10-1), we have the final result (denoted as Eq.(10-2)):
>
> $$
> \theta_{-j}^{[t+1]} - \theta^{[t+1]}= (\theta_{-j}^{[t]} - \theta^{[t]}) - \frac{\eta_t}{|S_t|} \sum_{i \in S_t\setminus\{j\}} (g(z_i; \theta_{-j}^{[t]}) -g(z_i; \theta^{[t]}))+\frac{\eta_t}{|S_t|}\mathbf{1}_{j \in S_t}g(z_i; \theta^{[t]})
> $$
>
> $$
> \approx (\theta_{-j}^{[t]} - \theta^{[t]}) - \frac{\eta_t}{|S_t|} (|S_t|  H^{[t]}(\theta_{-j}^{[t]} - \theta^{[t]}) )+\frac{\eta_t}{|S_t|}\mathbf{1}_{j \in S_t}g(z_i; \theta^{[t]})\
> $$
>
> $$
> = (\theta_{-j}^{[t]} - \theta^{[t]}) - \eta_t H^{[t]}(\theta_{-j}^{[t]} - \theta^{[t]}) +\frac{\eta_t}{|S_t|}\mathbf{1}_{j \in S_t}g(z_i; \theta^{[t]})
> $$
>
> $$
> = (I- \eta_t H^{[t]})(\theta_{-j}^{[t]} - \theta^{[t]}) +\frac{\eta_t}{|S_t|}\mathbf{1}_{j \in S_t}g(z_i; \theta^{[t]}).
> $$
> **This derivation confirms the correctness of Eq. (14), including the last term.**
>
>
>
> ## Concern 2:
>
>  Thank you for the thoughtful examination. We appreciate the opportunity to clarify and highlight our contributions.
>
> ### 1) Clarification of Influence Function Limitations
> As stated in our Introduction (lines 038-049), influence functions have three shortcomings:
> - **First**, they can only estimate a sample's overall influence on the final model, rather than its influence at different stages of training, which DIT can do.
> - **Second**, influence functions rely on optimal model parameters $ \theta^* $, assuming model optimality and convexity [4].
> - **Third**, high computational cost.
>
> [4] S. Basu, P. Pope, and S. Feizi, “**Influence Functions in Deep Learning Are Fragile**,” ICLR, 2021.
>
>
> ### 2) Response to Convexity Assumption
> The definition of influence function requires model optimality and convexity assumptions [4], according to "A. Deriving the Influence Function" by [Koh and Liang (2017)] (https://arxiv.org/pdf/1703.04730). They state:
>
> "We further assume that $ R $ is **twice-differentiable and strongly convex** in $ \theta $"
>
> "Since $ \hat{\theta}_{\epsilon,z} $ is a minimizer”
>
> “Since $ \hat{\theta} $ minimizes $ R $, we have **$ \nabla R(\hat{\theta}) = 0 $**.”
>
> Papers [1], [3] still rely on these assumptions. Otherwise, influence functions cannot be established. For example, [3] requires:
> $$
> \theta^* := \arg \min_{\theta \in \Theta} [F(\theta) := \mathbb{E}_{Z \sim P}[\ell(Z,\theta)]]
> $$
> "$\ell$ is a loss function that is **closed, convex, and thrice continuously differentiable** in the second argument. We assume this argmin is unique."
>
> ### 3) Response to Computational Efficiency
> DIT can efficiently estimate the sample influence within any single epoch, requiring far less computational costs than traditional influence functions and their extension [1-3].
> Table 4 shows that DIT at the First, Mid, and Last Epochs, consistently outperforms traditional methods in accuracy, particularly in DNN and CNN. This highlights DIT's ability to achieve efficiency with high performance.
> In contrast, the reliance on top-p eigenvalues in Arnoldi iterations[1], can lead to significant inaccuracies.

---

> > ### Author Response · Authors · 2024-11-26
> >
> > Dear Reviewer,
> >
> > Thank you again for your thorough review and insightful comments that helped improve our work. As the discussion phase ends soon, we would like to know if you have any additional feedback on our response. If our response and new experiments address your concerns, we would appreciate your consideration in raising the score.
> >
> > Thank you for your time and consideration.
> >
> > Best regards,
> > Authors

---

> > ### Comment · Reviewer_5BaV · 2024-11-27
> > **Response to Authors**
> >
> > I thank the authors for their response.
> >
> > Concern 1.
> >
> > I thank the authors for the step-by-step derivation and please make this clear in the revision.
> >
> > Concern 2.
> > 1. The convexity assumption in [Koh and Liang (2017)] is only for convenience. For $\hat{\theta}$ to be the unique minimizer and $H_{\hat{\theta}}$ to be positive definite, it is enough to assume that $\ell$ is locally strong convex around its minimizer.
> > 1. The authors did not comment on "the idea of projection has already appeared in the Maximum Influence Subset framework [2]" and "boundedness of gradient and hessian are assumed instead, which are very strong. In [3], only tail assumptions are made, so it is unclear whether the authors' argument requires less stringent assumptions."
> >
> > I will keep my score.

---

> > > ### Author Response · Authors · 2024-11-28
> > >
> > > Concern 1: We appreciate acknowledging the correctness of our derivation. The detailed step-by-step proof is included in the Appendix.
> > >
> > > Concern 2:
> > >
> > > 1. **On Model Convergence:**
> > > DIT's fundamental innovation is measuring sample influence during *any* training period, regardless of convergence. This is critically different from influence functions which require local minimizers. For example, DIT can analyze:
> > > - Sample influence during the first epoch
> > > - Impact when escaping local optima
> > >
> > > These capabilities are impossible with influence functions, which are restricted to regions near local optima.
> > >
> > > 2. **On Projection Methods:**
> > > We thank the reviewer for his recognition of the projection method and and for citing [2]. DIT achieves two key functions that [2] cannot:
> > > - Arbitrary time windows analysis: Our analysis across arbitrary time windows during training, rather than being limited to convergence analysis or post-training. (Section 5)
> > > - Comprehensive projection queries: Our query vectors enable analysis of multiple model characteristics that [2] cannot address, including prediction changes ($q = \nabla_\theta f(x_{\text{test}}; \theta^{[t]})$), parameter importance ($q = e_i$), and gradient alignments ($q = \nabla_\theta \ell(z_j; \theta^{[t]})$). (Section 4.1 and Appendix A.3)
> > >
> > > 3. **On Assumptions:**
> > > The reviewer suggests that we make strong boundedness assumptions. However, [3] actually makes much stronger assumptions. [3] includes Bounded gradient (d), and Bounded Hessian (e).
> > > ---
> > >
> > > To ensure accuracy, **we quote directly from [3]:**
> > >
> > > "$\ell: \mathcal{Z} \times \Theta \to \mathbb{R}_+$ is a loss function that is closed, convex, and thrice continuously differentiable."
> > >
> > > "Assumptions. We make the following assumptions.
> > >
> > > (a) $\|D^3_\theta\ell(z, \theta)[u, u, v]\| \leq$ $R \|u\|^2_{\nabla^2 \ell(z,\theta)}$ $ \|v\|_2$,
> > >
> > > for $R \geq 1$ where $ D^3_xf(x)[u, u, v] := \frac{d}{dt}\langle u, \nabla^2f(x + tv) u\rangle|_{t=0}$ for $f$ thrice continuously differentiable.
> > >
> > > (b) There exists $K_1 \geq 1$ such that the normalized gradient $H_{\*}^{-1/2}\nabla\ell(Z, \theta_\*)$ at $\theta_*$ is sub-Gaussian with parameter $K_1$"
> > >
> > > (c)  There exists $K_2 \geq 1$ such that the standardized Hessian $H(\theta_*)^{-1/2}\nabla^2\ell(Z, \theta_*)H(\theta_*)^{-1/2} - I_p$ at $\theta_*$ satisfies a Bernstein condition with parameter $K_2$. Moreover,
> > > $\sigma^2_H := \left\|V\left(H(\theta_*)^{-1/2}\nabla^2\ell(Z, \theta_*)H(\theta_*)^{-1/2}\right)\right\|_2$
> > > is finite, where we denote $V(H) = \mathbb{E}[HH^\top] - \mathbb{E}[H]\mathbb{E}[H]^\top$ for a random matrix $H$.
> > >
> > > (d) **Bounded gradient**:The normalized gradient is bounded in a neighborhood of $\theta_\*$, i.e., there exist $M_1 \geq 1, \rho \in (0, R^{-1}]$ such that
> > > $\|\nabla \ell(z, \theta)\| \leq M_1$ for all $z \in \mathcal{Z}$ and $\|\theta-\theta_{\*}\|_{H*} \leq \rho$
> > >
> > > (e) **Bounded Hessian** :The normalized Hessian is bounded in a neighborhood of $\theta_*$, i.e., there exist $M_2 \geq 1, \rho \in (0, R^{-1}]$ such that  $\|H(z, \theta)\| \leq M_2$ for all $z \in \mathcal{Z}$ and $\|\theta - \theta_*\|\leq \rho$
> > >
> > > (f) **Bounded test function**:The test function $h$ is bounded as $\|\nabla h(\theta)\| \leq M_1'$ and $\|H\*^{-1/2}\nabla^2h(\theta)H\*^{-1/2}\| \leq M_2'$   for all $\|\theta - \theta_*\|  \leq \rho$."
> > >
> > > ---
> > >
> > >
> > > 4. **On Tail Assumptions:**
> > > The reviewer states that "In [3], only tail assumptions are made." However, as evidenced by the direct quotes above, [3] makes multiple strong assumptions beyond tail conditions, including convexity, thrice continuous differentiability, sub-Gaussian gradients, and Bernstein conditions. In contrast, DIT's assumptions are looser and more applicable.

---

### Official Review · Reviewer_MuUJ · 2024-11-03

**Soundness:** 3
**Presentation:** 3
**Contribution:** 3
**Rating:** 8
**Confidence:** 3

**Summary:**

Dear Authors, thanks for submitting to ICLR.

In this paper, the author introduce a novel method for monitoring learning dynamics, termed DIT, which is designed for application across arbitrary neural networks trained with stochastic gradient descent (SGD). Unlike traditional influence functions and their extensions, DIT is not restricted to convex scenarios, demonstrating enhanced effectiveness in common non-convex cases. Additionally, DIT is computationally efficient, avoiding large matrix multiplications and thus enabling its application to complex models. Importantly, the implementation of DIT integrates seamlessly with standard SGD without requiring any modifications, making it readily applicable to a broad range of existing models through standard training routines.

**Strengths:**

1. Compared with conventional methods like influence function and its extension, the DIT can monitor the influence of whole training process, instead of solely showing the influence on final loss.
2. On the computational side, this method avoids large matrix multiplications, making it highly efficient. This efficiency is particularly critical for complex models where computational overhead is a key concern. In cases where computation is less of an issue, a leave-one-out test could be performed, but for large-scale or resource-intensive models, the lightweight nature of this approach makes it especially advantageous.
3. Overall, DIT outperform IF in terms of accruacy, showing very significant correlation with LOO test.
4. The paper is well-structured and presented in a clear, logical manner, with broadly sound reasoning throughout.

**Weaknesses:**

1. The evaluation is not comprehensive enough. So far only simple model like CNN and DNN are included. This limits understanding of DIT's performance on more advanced tasks. For instance, it remains unclear if DIT can be effectively applied to fine-tuning large language models (LLMs).
2. Need more detailed result about the computation overhead. So far the experiment is done on a 8 GPU server, and the LOO can be used for these tasks (CNN, DNN, LR etc.). Quantify the computing overhead will help the users to implement DIT properly.
3. In the abstract, the author mentions curriculum learning (CL); however, the paper lacks a direct use case demonstrating how DIT contributes to CL. I see the results in Table 2, but how it will guide a CL process is unclear.
(Please see more detailed suggestions in Questions )

**Questions:**

There are few detailed questions I would like to get more clarity.

1. For equation (1), line 080, page 2, is it  a critical condition to the DIT? For instance, if I use certain realy stopping method and the equation (1) is not minimized, would DIT still work? Suggestion: add a test for DIT when the model is fully converged.
2. Will this method helps the fine tunning of large pretrained models like LLM? Specifically, how does DIT handle new training samples for a pre-trained model, and does it require retraining the entire model from scratch? Suggestion: at least a clarification about the limitation would be helpful after the evluation, section 5. If the model is too large, maybe investigate a pretrained module inside would be good enough.
3. Please quatify the cumputation efficicency gain of DIT. The result includes but not limited to: computation time compared with LOO, memory consumption. This will highlight the efficiency gain of DIT. This content can appear in section 5.
4. Please add a use case for curriculum learning (CL) in section 5.5, as it is mentioned in the abstract. It's currently unclear how DIT manages dynamic training data, a common feature in CL setups. It is also not clear the correlation showed in Table 2 is sufficient without looking into a real CL task.
5. Given its potential as a highly useful tool, will DIT be made open-source upon acceptance?

**Details Of Ethics Concerns:**

No such concern

---

> ### Author Response · Authors · 2024-11-19
> **DIT does NOT rely on model convergence or global optimality, and well-suited for fine-tuning LLMs**
>
> ## 1. Response to Model Optimality
>
> Thank you for this insightful question. **DIT does NOT rely on model convergence or global optimality**, so equation (1) is not a critical condition for DIT. In contrast, Influence Functions (IF) require model convergence and global optimality [1,2]. Section 5.3 compared DIT, IF, and LOO, all of which fully converge.
>
> 1)	Section 5.5 (Table 4) has shown DIT advantage by using DIT in the first epoch and middle epoch, when neither converges.
>
> ** Table 4: Number of correctly identified flipped samples **
> | Flipped | Model | IF              | Full DIT        | LOO             | First Epoch      | Mid Epoch       | Last Epoch      |
> |---------|-------|-----------------|-----------------|-----------------|------------------|-----------------|-----------------|
> | 5%      | DNN   | **2.94 ± 2.01** | 9.06 ± 1.85     | 8.81 ± 1.98     | **8.25 ± 2.33** | **8.88 ± 2.09 ** | 9.38 ± 1.98     |
> | 5%      | CNN   | **5.88 ± 2.26** | 10.50 ± 1.32    | 10.44 ± 1.32    | **8.75 ± 2.11** | **10.69 ± 1.16 **| 11.06 ± 1.32    |
> | 10%     | DNN   | **7.50 ± 3.34** | 20.75 ± 3.01    | 19.94 ± 3.77    | **20.31 ± 2.78** | **20.50 ± 3.22 ** | 21.31 ± 3.77    |
> | 10%     | CNN   | **15.00 ± 2.83**| 21.81 ± 3.11    | 21.75 ± 3.11    | **18.44 ± 4.37** | **22.19 ± 2.81 ** | 23.56 ± 3.11    |
>
> For example, in MNIST-CNN experiments with 5% flipped labels, DIT First Epoch achieves (8.25) achieves 2.8x better accuracy** than IF (2.94). DIT Mid Epoch achieves (8.88) achieves 3.0x better accuracy** than IF (2.94). Similar superior performance has been observed in other settings.
>
> 2) Section 5.4 provides a comprehensive analysis of DIT's performance across early, middle, late, and full training stages (full training is converged), showing consistent effectiveness throughout the training process.
> 3) Our theoretical analysis in Section 3.3 does not assume and rely on model convergence.
>
> [1] Koh and Liang. "Understanding black-box predictions via influence functions." ICML 2017.
>
> [2] Basu et al. "Influence functions in deep learning are fragile." ICLR 2021.
>
>
> ## 2. Response to Suitable for Large Pre-trained Models
>
> YES. DIT is well-suited for fine-tuning LLMs.
>
> 1) Our method requires **no assumptions about global optimality or convergence**, making it ideal for analyzing fine-tuning. DIT naturally handles fine-tuning by treating pre-trained model's parameters as the initial state $\theta_0$. No need to retrain. DIT can analyze the sample influence during fine-tuning.
>
> 2) DIT is model-agnostic and scales to any architecture size. Whether fine-tuning the entire model or specific modules like adapters, our query-based approach naturally tracks the influence on updated parameters.
>
> 3) Our empirical results in Tables 3 and 4 show that DIT can accurately identify influential samples in early training stages. This early detection capability means DIT can effectively analyze LLM fine-tuning while keeping storage costs minimal, as we only need to track the early stage.
>
> We have added a detailed discussion of DIT's practical considerations in the Appendix.

---

> ### Author Response · Authors · 2024-11-19
>
> ## 3. Response to Performance Quantification
>
> We appreciate the reviewer's constructive suggestion. We have compared both theoretical complexity analysis and empirical evaluation in Table 6 and added them in the Appendix.
>
> Table 6. Comparison of Different Sample Influence Analysis Methods
> | Aspect                          | DIT                               | First-Epoch DIT | Influence Functions | Leave-One-Out  |
> |---------------------------------|-----------------------------------|-----------------|--------------------------|---------------------|
> | Time Complexity     | $O(T\|S_t\|p)$            | $O(E\|S_t\|p)$           | $O(p^3 + Np^2)$       | $O(NT\|S_t\|p)$        |
> | Storage Complexity  | $O(T(\|S_t\| + p))$       | $O(E(\|S_t\| + p))$      | $O(p^2)$               | $O(p)$               |
> | Robustness to Non-convergce | Yes                               | Yes         | No                       | NO                 |
> | Robustness to Non-convexity | Yes                               | Yes         | No                       | Yes                 |
> | Robustness to Global optimality| Yes                               | Yes         | No                       | Yes                 |
> | Adaptability              | Flexible   | Flexible         | Static                   | Static              |
>
> *Note: T = total steps, E = steps per epoch, p = parameters, $\|S_t\|$ = batch size*
>
>
> DIT shows significant efficiency gains in empirical evaluation using a DNN model on 256 randomly selected MNIST samples. First-Epoch DIT completes in 0.1 seconds, while Full DIT takes 17.43 seconds, compared to IF's 10.81 seconds and LOO's 63.88 seconds. As shown in Table 4, analyzing just the first epoch allows DIT to achieve 2.8x better accuracy than IF for DNNs (8.25 vs 2.94) and 1.5x better accuracy for CNNs (8.75 vs 5.88).
>
> While Full DIT is slightly slower than IF, it offers flexibility in the analysis window and robustness to model non-convexity, non-convergence, and global optimality. The First-Epoch DIT variant achieves a 638× speedup compared to LOO while maintaining high accuracy (Table 4), making it particularly suitable for real-time applications in deep learning scenarios.
>
>
> ## 4. Response to Curriculum Learning
>
> Thank you for your comment about Curriculum Learning (CL). Our experimental results, particularly the influence pattern analysis and cross-training stage similarity study, already provide strong evidence for DIT's applicability to CL:
>
> 1) Section 5.2 identifies four distinct influence patterns (Stable, Early, Late Blooming, and Highly Fluctuating). This enables DIT to serve as an ideal data preprocessing method for difficulty categorization and optimal sample ordering, a key requirement for CL. Specifically,
> - Early Influencers are candidates for the initial curriculum stages due to their significant early-stage impact.
> - Late Bloomers align with curriculum progression by being more effective in later stages.
> - Stable Influencers can serve as anchor samples across different stages.
>
> This pattern-based categorization provides an efficient and effective data preprocessing method for CL **without manual labeling**.
>
> 2) Our training stage analysis (Section 5.4, Table 2) reveals:
> - High correlation between early training and full training influence (e.g., 0.91 ± 0.02 for 20News-LR) enables early identification of important samples.
> - Low correlation between early and late stages (e.g., 0.18±0.21 for MNIST-DNN) confirms the dynamic evolution of sample importance, supporting the necessity of progressive learning.
>
> These findings provide a theoretical foundation for dynamic curriculum design.
>
>
> ## 5. Response to Open-source DIT
>
> I appreciate your interest in DIT's accessibility. DIT is already open-source at https://github.com/dynamic-infl-tracker/DIT and will remain so regardless of acceptance. We are committed to maintaining its open-source status regardless of the paper's acceptance outcome. Detailed experimental settings and reproducibility guidelines can be found in Section 5.1 and the repository. We welcome further discussion and collaboration.

---

> > ### Comment · Reviewer_MuUJ · 2024-11-26
> >
> > Thanks for your detailed reply.
> >
> >  I think all my major concerin is resolved. I will keep my score wish for the best of the paper.

---

### Official Review · Reviewer_DLNR · 2024-11-04

**Soundness:** 2
**Presentation:** 3
**Contribution:** 2
**Rating:** 3
**Confidence:** 3

**Summary:**

This paper introduces the Dynamic Influence Tracker (DIT), a novel approach to estimate the dynamic influence of individual training samples in models trained with SGD. Unlike traditional influence methods that provide only static influence estimates, DIT tracks influence over arbitrary time windows during training. DIT operates through a two-phase algorithm: the training phase, which captures and stores relevant information about the model’s parameter trajectory, and the inference phase, which uses this information to estimate sample influence within specific time frames. The authors derive a theoretical error bound for their estimator, demonstrating its applicability in non-convex settings, and present extensive empirical evaluations that highlight how sample influence evolves during training. The experiments show that DIT outperforms existing methods in accuracy.

**Strengths:**

The example provided in Section 5.2 illustrates that sample influence changes dynamically over the course of training, effectively supporting the motivation for DIT.

**Weaknesses:**

1.	The theoretical bound in Equation (18) grows exponentially with t making it vacuous for long training intervals.

2.	The empirical evaluations primarily use simple models, like two-layer CNNs, and small datasets, such as MNIST, which limits the demonstration of DIT's scalability and applicability to more complex scenarios.

3.	The empirical results do not clearly show the practical utility of a dynamic influence function. In Section 5.5, the improvement from using DIT is marginal, and the correlations in Section 5.4 lack comparison with baseline methods. Although Figure 2 shows that DIT outperforms Influence Functions (IF) when using the full-time window, DIT’s approach comes with a high storage requirement, which could limit its scalability.

4.	The proposed algorithm relies on a Taylor approximation, requiring storage of all intermediate models during SGD, which results in a trade-off between time and memory complexity that could be problematic for large-scale models.

**Questions:**

Could the authors present a compelling application for the proposed DIT approach that justifies the substantial storage required for retaining all intermediate models and SGD batches throughout training?

---

> ### Author Response · Authors · 2024-11-14
> **Storage is **optional**. DIT one epoch  (storage ~6MB) achieves much better ($\times$3) accuracy than baselines**
>
> ## Response to Questions about storage costs.
>
> **Storing intermediate models is optional**. DIT can analyze any time window $[t_1,t_2]$. Table 4 shows **one epoch**  (storage ~6MB) DIT achieves superior performance. In detecting flipped labels task in Sec. 5.5, using only the first epoch, DIT significantly outperforms existing methods: achieving 2.8x better accuracy than IF for DNNs (8.25 vs 2.94) and 1.5x better accuracy for CNNs (8.75 vs 5.88).
>
> ** Table 4: Number of correctly identified flipped samples **
> | Flipped | Model | IF              | Full DIT        | LOO             | First Epoch      | Mid Epoch       | Last Epoch      |
> |---------|-------|-----------------|-----------------|-----------------|------------------|-----------------|-----------------|
> | 5%      | DNN   | **2.94 ± 2.01** | 9.06 ± 1.85     | 8.81 ± 1.98     | **8.25 ± 2.33**  | 8.88 ± 2.09     | 9.38 ± 1.98     |
> | 5%      | CNN   | **5.88 ± 2.26** | 10.50 ± 1.32    | 10.44 ± 1.32    | **8.75 ± 2.11**  | 10.69 ± 1.16    | 11.06 ± 1.32    |
> | 10%     | DNN   | **7.50 ± 3.34** | 20.75 ± 3.01    | 19.94 ± 3.77    | **20.31 ± 2.78** | 20.50 ± 3.22    | 21.31 ± 3.77    |
> | 10%     | CNN   | **15.00 ± 2.83**| 21.81 ± 3.11    | 21.75 ± 3.11    | **18.44 ± 4.37** | 22.19 ± 2.81    | 23.56 ± 3.11    |
>
> ## Response to Comment 1: "The theoretical bound in Equation (18)."
>
> We appreciate this insightful observation, and we would like to provide important clarifications.
>
> 1)  **DIT achieves NEAR-ZERO error empirically**; the bound is a worst-case mathematical guarantee. Experimental results (Table 2) show that DIT nearly matches the ground truth LOO: 0.99±0.01 in LR-Adult, 0.99±0.01 in LR-20News, and 0.95±0.02 in DNN-Adult, **showing near-zero error**.
>
> 2) **Existing methods provide NO GUARANTEE for non-converged non-convex settings.**  Our bound, despite its exponential growth in the worst case, represents an advancement over existing methods.
>
> 3) DIT can analyze any time window $[t_1,t_2]$, which means a small upper error bound with small $t_1, t_2$.
>
>
> ## Response to Comment 2: "The empirical evaluations primarily use simple models."
> It was deliberate for several reasons:
>
> 1) **Limited by Baseline Methods**. To validate DIT, we must compare with existing methods like Influence Functions (IF) and Leave-One-Out (LOO), which are prohibitively expensive for larger models:
>    - LOO requires O(N) complete model retraining cycles. With N=50,000 (MNIST size) and hours per training cycle, this would **TAKE YEARS** of computation!
>    - IF requires computing the inverse Hessian matrix (O(p²) complexity) and is limited by scalability, requiring model convergence and global optimality.
>
> 2) Our results show DIT's advantages increase with model complexity (LR < DNN < CNN), suggesting even stronger benefits in more complex settings.
>
> 3) DIT is model agnostic with no architectural assumptions, provides error bounds without convexity requirements. We welcome future work on larger architectures through our open-source implementation.
>
> ## Response to Comment 3 about the empirical results.
>
> 1) The improvements from DIT are **significant**, especially in complex models using the last epoch of DIT in Section 5.5:
>    - **DIT (9.38) achieves 3.2x better accuracy** than IF (2.94), for DNNs with 5% flipped labels.
>    - **DIT (11.06) achieves 1.9x better accuracy** than IF (5.88), for CNNs with 5% flipped labels.
>
> 2) Section 5.4's correlation analysis demonstrates a unique capability of DIT: tracking how sample importance evolves during training. This is not a comparison with baselines because **existing methods CANNOT perform this temporal analysis at all**. They only provide static, end-of-training measurements.
>
> 3) Storage is optinal. DIT can achieve superior performance using only first-epoch data (Table 4).  In contrast, the baseline IF is limited by scalability (requires model convergence and global optimality) rather than our scheme DIT.
>
> ## Response to Comment 4 about storage.
> **NO NEED to store all intermediate models**. Using only first-epoch data (only about 6MB), DIT still significantly outperforms IF: achieving 2.8x improvement for DNNs (8.25 vs 2.94) and 1.5x improvement for CNNs (8.75 vs 5.88). This demonstrates that full storage is optional and not required. DIT also offers computational advantages by avoiding both the O(N) complete model retraining required by LOO and the expensive Hessian matrix inversion needed by IF.

---

> > ### Author Response · Authors · 2024-11-26
> >
> > Dear Reviewer,
> >
> > Thank you again for your thorough review and insightful comments that helped improve our work. As the discussion phase ends soon, we would like to know if you have any additional feedback on our response. If our response and new experiments address your concerns, we would appreciate your consideration in raising the score.
> >
> > Thank you for your time and consideration.
> >
> > Best regards,
> > Authors

---

> > > ### Comment · Reviewer_DLNR · 2024-12-02
> > >
> > > Thank you to the reviewer for preparing the response. It's encouraging to see that the improvement doesn't require storing all intermediate models. However, I am still not fully convinced by the other concerns addressed in the response, particularly regarding the theoretical results and experiments with more complex models, which may require significant revisions. Therefore, I will maintain my current score.

---

> > > > ### Author Response · Authors · 2024-12-02
> > > >
> > > > Dear Reviewer,
> > > >
> > > > Thank you for your continued feedback. We would like to address your remaining concerns:
> > > >
> > > > 1. Regarding the exponential error bound:
> > > >    - This is due to our **minimal assumptions about model convergence**. The exponential form reflects the worst-case error accumulation inherent in model training (e.g., if one-step error is $a$, accumulated error grows as power of $a$), **rather than a limitation of our estimation method**.
> > > >    - Our empirical results show that **DIT achieves near-zero error in practice**, significantly outperforming existing approaches.
> > > >
> > > > 2. On scalability:
> > > >    - Our study represents large-scale evaluations to date for measuring "*the impact of including vs. excluding individual training points*" due to the inherent computational complexity of this task.
> > > >    - We will include additional experiments with larger models where **baseline methods become computationally infeasible**, to further demonstrate DIT's unique capability in scaling to complex architectures.
> > > >
> > > > Best regards,
> > > > Authors

---

> > > > > ### Author Response · Authors · 2024-12-04
> > > > > ****Additional experiment**: One-Epoch DIT Achieves >98% Detection Rate on Large-Scale ViT/ImageNet (86M Parameters)**
> > > > >
> > > > > Dear Reviewer:
> > > > >
> > > > > We conducted additional experiments on Vision Transformer ViT-B/16 trained on ImageNet-1K. This represents a state-of-the-art complex setting with:
> > > > > - **ViT-B/16 model scale**: **86M** parameters, 12 transformer layers, 768 hidden dimensions, 12 attention heads.
> > > > > - **ImageNet-1K dataset scale**: 1.28M images, 1000 classes, resolution 224×224.
> > > > >
> > > > > We evaluated DIT's ability to identify malicious samples by Flipped Label Sample Detection. Specifically, we:
> > > > > 1. Flipped X samples from class "brambling" (n01530575) to "daisy" (n11939491), $X \in \{50,100,150,200,250,300\}$
> > > > > 2. Randomly sampled 1000-X benign training samples
> > > > > 3. Applied DIT to analyze these 1000 samples' influence using **only one epoch** SGD information
> > > > > 4. Examined how many samples with the flipped labels are included in the X samples with the lowest DIT scores
> > > > >
> > > > > Results of identifying flipped label samples using one-epoch DIT:
> > > > >
> > > > > Table: Flipped Label Detection Results using **One-Epoch DIT (ViT-B/16 on ImageNet-1K)**
> > > > >
> > > > > | Number of Flipped Labels (X) | Number of Detected | Detection Rate |
> > > > > |:---------------------------:|:-----------------:|:--------------:|
> > > > > | 50                         | 41          | 82.00%         |
> > > > > | 100                         |98            | 98.00%         |
> > > > > | 150                         |148           | 98.67%         |
> > > > > | 200                        | 199            | 99.50%         |
> > > > > |250                         | 247            | 98.80%         |
> > > > > |300                         | 299            | 99.67%         |
> > > > >
> > > > > These results demonstrate that: DIT **scales effectively** to modern architectures (86M parameters) and maintains **consistently high accuracy** (>98% for most cases).
> > > > >
> > > > > This comprehensive evaluation on ViT/ImageNet directly addresses your concerns about scalability and practical utility. We will include these results in the revised paper.
> > > > >
> > > > > Best regards,
> > > > >
> > > > > Authors

---

### Official Review · Reviewer_XkZ2 · 2024-11-04

**Soundness:** 3
**Presentation:** 3
**Contribution:** 4
**Rating:** 6
**Confidence:** 3

**Summary:**

Understanding sample influence is an important task that has multiple different use tasks. However, existing estimators only provide a single static assessment. The authors propose a novel way that can dynamically and efficiently estimate sample influence across different stages. They provide theoretical bounds and the empirical evaluation shows a much stronger performance than the baseline.

**Strengths:**

1. The paper addresses an important problem. While providing a static assessment of sample influence is already useful, providing dynamic assessment potentially enables many interesting usages such as pinpointing why the model performance degrades or certain unwanted behavior appears after a certain training period.

2. The empirical performance is strong. The correlation with the ground truth is often pretty high (> 0.9), which is much higher than the influence function baseline.

**Weaknesses:**

1. It is unclear whether the error bound is meaningful as it grows exponentially. If the exponential growth is inevitable for the non-convex case, then it seems better to show a stronger bound for the convex case instead.

**Questions:**

1. While the authors already show the Kendall’s Tau correlations across training stages, it would be of the interest of the readers to see the correlation with respect to each of the evolution patterns: Stable Influencer, Early Influencers, Late Bloomers, and HighlyFluctuating. Since stable influencers are the majority, one can have high correlation by only predicting stable influencers accurately, while it might not correctly identify the Early Influencers and the Late Bloomers, which in some use cases is more of the interest.
2. Can the authors clarify under which condition they think the error bound could be meaningful? Also, it might be useful if the authors can show the error bound for the convex case if it can be much better than the non-convex case. Additionally, bounding this term seems a bit weird as it can be influenced by reparametrization (scaling the data by c in LR seems to result in a 1/c scaling in the parameter and the error bound).

---

> ### Author Response · Authors · 2024-11-15
> **New Experimental Results show DIT's Strong Performance Across **All** Evolution Patterns**
>
> ## 1. Response to Concern About Pattern-Specific Correlation Analysis
> Thank you for suggesting the analysis of correlation across different evolution patterns. We agree that high performance on only stable influencers could mask performance on others. We conducted additional experiments to analyze DIT's performance separately on each evolution pattern (Stable Influencer, Early Influencers, Late Bloomers, and Highly Fluctuating) in MNIST-DNN, following the process described in Section 5.2. The results are shown below:
>
> **Table: Performance comparison of DIT and IF for different types of data points using MNIST-DNN**
>
> | Dataset                          | Pearson DIT | Pearson IF | Spearman DIT | Spearman IF | Kendall's Tau DIT | Kendall's Tau IF | Jaccard DIT | Jaccard IF |
> |----------------------------------|-------------|------------|--------------|-------------|-------------------|------------------|-------------|------------|
> | **Stable Influencers**           | **0.95±0.03** | 0.23±0.39  | **0.96±0.03**  | 0.16±0.40   | **0.87±0.05**       | 0.13±0.28         | **0.82±0.12** | 0.26±0.16  |
> | **Early Influencers**            | **0.94±0.04** | 0.35±0.29  | **0.98±0.01**  | 0.35±0.30   | **0.92±0.03**       | 0.26±0.23         | **0.89±0.07** | 0.29±0.20  |
> | **Late Bloomers**                | **0.98±0.02** | 0.23±0.46  | **0.98±0.02**  | 0.19±0.38   | **0.90±0.05**       | 0.15±0.27         | **0.85±0.10** | 0.27±0.21  |
> | **Highly Fluctuating Influencers** | **0.76±0.18** | -0.10±0.54 | **0.72±0.18**  | -0.08±0.48  | **0.63±0.21**       | -0.09±0.40        | **0.52±0.34** | 0.15±0.17  |
>
> The results reveal several key findings:
> - **Our method DIT performs well across all pattern types**: Early Influencers (Kendall's Tau = 0.92±0.03) and Late Bloomers (0.90±0.05), Stable Influencers (0.87±0.05), validating its effectiveness across different influence patterns.
> - **DIT consistently outperforms IF** across all pattern types.
> - For Highly Fluctuating Influencers, both methods show decreased performance with higher variance, though DIT maintains positive correlation while IF shows negative correlation (Kendall's Tau: -0.09±0.40). This discrepancy arises because IF relies on static Hessian estimation at the optimal point, failing to capture dynamic changes during training. While DIT does not require this condition, its performance is still affected by the limitations of first-order Taylor approximation in capturing highly non-linear changes.
>
> Furthermore, the flipped label detection experiments (Table 4) show the detection effectiveness across **first epoch**, **mid epoch** and **last epoch**, indirectly showing DIT's capability to identify Early Influencers and Late Bloomers.
>
> We have included these findings in the revised manuscript and discussed their implications for practical applications. Thank you again for this valuable suggestion that has helped make our evaluation more comprehensive and insightful!

---

> ### Author Response · Authors · 2024-11-15
> **1. DIT does **not require model convergence**, resulting in exponential error bounds. 2. Model **performance (loss, prediction) remains unchanged** regardless of parameter scaling.**
>
> ## 2.1 Response to Concern About Meaningful Error Bounds
>
> Thank you for your attention to the exponential bound. Note that DIT is applicable to **non-converged** and **non-convex** models in any time window $[t_1, t_2]$. The exponential form emerges from the recursive nature of error propagation, where each SGD step compounds previous errors multiplicatively.
>
> 1) The bound is **meaningful for short time windows and small learning rates** (common in practice). We guarantee a much tighter upper error bound with small $\[t_1, t_2\]$, such as the first epoch. As shown in Table 4, analyzing just the first epoch allows DIT to achieve 2.8x better accuracy than IF for DNNs (8.25 vs 2.94) and 1.5x better accuracy for CNNs (8.75 vs 5.88).
>
> 2) **Existing methods provide NO GUARANTEE for non-converged non-convex settings.** Our bound, despite its exponential growth in the worst case, represents an advancement over existing methods.
>
> 3) While theoretical bounds provide worst-case guarantees, empirical evidence strongly supports DIT's practical effectiveness. Experimental results (Table 2) show that DIT nearly matches the ground truth LOO: 0.99±0.01 in LR-Adult, 0.99±0.01 in LR-20News, and 0.95±0.02 in DNN-Adult, **showing near-zero error**. This suggests the theory's primary value lies in revealing key parameter relationships and guiding implementation choices.
>
> ## 2.2 Response to Reviewer about error bound for the convex case
>
> Our primary contribution is developing a method that effectively handles **non-converged**and **non-convex** scenarios without requiring strong assumptions. The exponential form emerges from the recursive nature of error propagation, where each SGD step compounds previous errors multiplicatively. Though our bound appears conservative, it reveals crucial parameter relationships and guides implementation without assuming convergency and convexity. Experimental results validated near-zero error performance.
>
>
> ## 2.3 Response to Concern About Scale Invariance of Error Bounds
>
> Thank you for your insightful observation regarding the scale dependence of our error bound. We appreciate this careful examination. We would like to clarify several important points:
>
> 1. **Left-right consistency**:
>    When the input data is scaled by $ c $ ($ X \rightarrow cX $), both sides of our bound scale consistently by $ 1/c $:
>    - The left-hand side $ \| \Delta \theta_{-j}^{[t_1,t_2]} - \hat{\Delta \theta}_{-j}^{[t_1,t_2]} \| $ scales by $ 1/c $ due to the natural scaling of parameter differences
>    - The right-hand side $ \frac{\tilde{B}}{M_H} (\exp(M_H \eta_{\max} (t_2+1)) + \exp(M_H \eta_{\max} (t_1+1)) - 2) $ also scales by $ 1/c $ due to the balanced scaling of its components ($ \tilde{B} \rightarrow c \cdot \tilde{B}, M_H \rightarrow c^2 \cdot M_H, \eta_{\max} \rightarrow \eta_{\max}/c^2 $)
>
> The observed scaling behavior reflects the **inherent properties of parameter space transformations** rather than a limitation of our analysis. In linear regression, when inputs are scaled by $ c $, the optimal parameters naturally scale by $ 1/c $ to maintain the same predictions.
>
> 2. **Invariance of Loss**: When considering our Query-based DIT, the actual influence on model behavior remains invariant under scaling. For example, with loss queries $ q(t) = \nabla_\theta \ell(z_{\text{test}}; \theta^{[t]}) $:
>
> $$
>    Q_{-j}^{[t_1,t_2]}(q) = \langle q(t_2), \Delta\theta_{-j}^{[t_2]} \rangle - \langle q(t_1), \Delta\theta_{-j}^{[t_1]} \rangle
> $$
>
>    While $ q(t) $ scales by $ c $ and $ \Delta \theta_{-j}^{[t]} $ scales by $ 1/c $, their **inner product remains unchanged**, showing that the practical influence on model behavior, i.e. **loss, is scale-invariant**.
>
> This analysis demonstrates that while the parameter-space measurements do scale with data transformation, the bound remains mathematically consistent and maintains its practical utility in capturing the influence of training samples on model behavior.
>
> Thank you again for this valuable comment, which has helped us clarify an important aspect of our theoretical analysis.

---

> > ### Author Response · Authors · 2024-11-26
> >
> > Dear Reviewer,
> >
> > Thank you again for your thorough review and insightful comments that helped improve our work. As the discussion phase ends soon, we would like to know if you have any additional feedback on our response. If our response and new experiments address your concerns, we would appreciate your consideration in raising the score.
> >
> > Thank you for your time and consideration.
> >
> > Best regards,
> > Authors

---

> > > ### Comment · Reviewer_XkZ2 · 2024-11-27
> > >
> > > Hi,
> > >
> > > I have read the rebuttal response and I raised my score to 6.
> > >
> > > Sincerely yours,
> > > Reviewer XkZ2

---

### Official Review · Reviewer_RUYz · 2024-11-04

**Soundness:** 3
**Presentation:** 3
**Contribution:** 2
**Rating:** 5
**Confidence:** 3

**Summary:**

The authors introduce the Dynamic Influence Tracker (DIT), a novel approach for estimating the influence of individual training samples within specific time windows in models trained via stochastic gradient descent (SGD). Unlike traditional methods, which provide static influence estimates, DIT dynamically tracks how the influence of each sample varies throughout training. The method comprises a two-phase algorithm: a training phase that stores information about the SGD process and an inference phase that computes the influence within a specified window. Experimental results across various datasets and models demonstrate DIT’s advantages in both accuracy and robustness over existing methods.

**Strengths:**

1. DIT addresses limitations in current methods by providing dynamic estimates, which allow for a more nuanced understanding of sample influence as training progresses. The authors also provide theoretical guarantees for the error bounds, ensuring the model is reliable, even in non-convex landscapes.

2. The experiments cover diverse datasets and various model architectures. The results demonstrate the advantages of DIT over existing methods in terms of accuracy and robustness.

3. DIT reduces computational overhead by using Hessian-vector approximations instead of directly computing or storing the Hessian, making it feasible for large-scale models and datasets.

**Weaknesses:**

1. It seems that the proposed method can only deal with models trained with SGD. Considering that many models are trained with more complex optimization methods such as Adam, the application of the proposed method seems somewhat limited.

2. It’s unclear whether the DIT is sensitive to hyperparameters, especially the learning rate and batch size, which can vary widely in real-world settings. Further analysis of these factors would strengthen the applicability of the method.

3. The error bound analysis relies on many assumptions (A1-A7) that look strong and may not hold in practice, such as the behavior of Hessians in deep networks.

4. It seems that the space and time complexity of the proposed method could still be challenging for very large-scale datasets and deep models.

**Questions:**

See Weakness.

---

> ### Author Response · Authors · 2024-11-14
> **DIT is a general framework being extended to other optimizers. By using just **single-epoch** analysis, DIT achieved better accuracy than baselines (8.25 vs IF's 2.94 in Table 4) with lower computational costs, making it practical for large-scale models.**
>
> ## Response to Concern 1 about SGD-trained Model:
>
> We appreciate this thoughtful comment and would like to clarify three important points:
>
> 1. **Significant advancement over existing methods**: Existing influence estimation methods are limited to a certain model (such as KNN [1]), or a model that can achieve global optimality, or has convex loss [2].  In contrast, DIT enables influence estimation for any model trained with SGD, regardless of type, convexity, or optimality. This represents a great theoretical and practical advancement in understanding how individual samples affect model training.
>
> 2. **Foundation of optimizers**: SGD is the fundamental building block of modern optimization algorithms. We present our method using SGD for clarity and accessibility. Understanding influence dynamics in SGD provides crucial insights that generalize to more complex optimizers.
>
> 3. **Extension to other optimizers**: DIT can be adapted to optimizers like Adam by slightly modifying parameter change tracking to account for momentum and adaptive rates, adjusting the Hessian approximation, and using the same influence computation framework. The core principles of DIT remain valid. We are actively working on explicit extensions to commonly used optimizers, with promising preliminary results.
>
> [1] R. Jia et al., “Efficient task-specific data valuation for nearest neighbor algorithms,” Proceedings of the VLDB Endowment, vol. 12, no. 11, pp. 1610–1623, 2018.
>
> [2] P. W. Koh and P. Liang, “Understanding black-box predictions via influence functions,” in ICML, 2017, pp. 1885–1894
>
>
> ## Response to Concern 2 Hyperparameter Sensitivity:
>
> The paper actually addresses hyperparameter sensitivity in several ways, though we could make this more explicit:
>
> 1. Learning Rate \& Batch Size Consideration:
> From Section 3's theoretical analysis, DIT incorporates learning rate $\eta_t$, and batch size $|S_{t}|$ in formulations:
> $Z_t = I - \eta_{t} H^{[t]}$, $ \mathbf{\tilde{1}}\_j^{[t]} = \mathbf{1}\_{j \in S_{t}}\frac{\eta_{t}}{|S_{t}|} g(z_j; \theta^{[t]})$, and
>
> $$
> \Delta \theta_{-j}^{[t_1,t_2]} \approx  \left(\prod_{k=t_1}^{t_2-1} Z_k - I\right)\left(\sum_{t=0}^{t_1-1} \left(\prod_{k=t+1}^{t_1-1} Z_k\right) \mathbf{\tilde{1}}\_j^{[t]}\right) \nonumber + \sum_{t=t_1}^{t_2-1} \left(\prod_{k=t+1}^{t_2-1} Z_k\right) \mathbf{\tilde{1}}\_j^{[t]}.
> $$
>
> 2. Experimental validation: The paper demonstrates robustness across different settings through testing on multiple datasets (Adult, 20Newsgroups, MNIST, EMNIST), model architectures (LR, DNN, CNN), and different noise levels (5%, 10%, 15%, 20%). The standard deviations reported in Tables 2 and 5 demonstrate stability across different initializations and configurations (16 runs with different random seeds).
>
> ## Response to Concern 3 about Assumptions (A1-A7):
>
> While the theoretical assumptions may appear strong, they serve as sufficient conditions for analysis, **not necessary** for practical effectiveness. Our assumptions are actually milder than previous work, as we only require Lipschitz continuity rather than the convexity assumed in influence functions.
>
> Moreover, our comprehensive empirical results directly address this concern. As shown in Table 2, on MNIST with DNNs, DIT achieves 0.90±0.07 Pearson correlation with ground truth, far exceeding IF's 0.25±0.28, despite potential violations of Hessian assumptions in deep networks.  DIT achieves reliable estimation even with approximate satisfaction of theoretical conditions, showing DIT's robust practical effectiveness beyond theoretical constraints.
>
> ## Response to Concern 4 about Scalability:
>
> Thank you for raising this important concern about scalability. Training large-scale models is computationally intensive, and measuring sample influence poses additional challenges. For a middle-sized model on MNIST, each full training run takes **several hours**. With MNIST's training set size of 50,000 samples, the traditional Leave-One-Out (LOO) method requires 50,000 full training runs, amounting to **years of computation time**. Influence Functions (IF) require expensive Hessian matrix inversions. Our method avoids both the extensive retraining and the expensive Hessian matrix inversions while maintaining high accuracy in estimation.
>
> More importantly, the storage and computation costs are **optional**.  DIT achieves superior performance using only **single-epoch** analysis. As shown in Table 4, analyzing just the **first epoch** allows DIT to achieve 2.8x better accuracy than IF for DNNs (8.25 vs 2.94) and 1.5x better accuracy for CNNs (8.75 vs 5.88). This single-epoch strategy drastically reduces both storage and computation time requirements while outperforming existing methods.
>
> ** Table 4: Number of correctly identified flipped samples (5% noise) **
> | Model | IF | **First Epoch** |
> |-------|-----------------|-----------------|
> | DNN | 2.94 ± 2.01 | **8.25 ± 2.33** |
> | CNN | 5.88 ± 2.26 | **8.75 ± 2.11** |

---

> > ### Author Response · Authors · 2024-11-26
> >
> > Dear Reviewer,
> >
> > Thank you again for your thorough review and insightful comments that helped improve our work. As the discussion phase ends soon, we would like to know if you have any additional feedback on our response. If our response and new experiments address your concerns, we would appreciate your consideration in raising the score.
> >
> > Thank you for your time and consideration.
> >
> > Best regards,
> > Authors

---

### Author Response · Authors · 2024-11-23
**Rebuttal Summary**

**Response to Reviewers**

We sincerely thank all reviewers for their valuable feedback. We summarize modifications and the new experiments made in our revised paper.


---

**Key Clarifications and New Results**

1. **Storage is selectable**: Storing and using only first-epoch data (~6MB), DIT significantly outperforms existing baseline methods:
   - 2.8x better accuracy than Influence function (IF) for DNNs (8.25 vs 2.94)
   - 1.5x better accuracy than IF for CNNs (8.75 vs 5.88)

2. **Comprehensive Pattern Analysis**: New experiments show DIT's effectiveness across all influence patterns:
   - Early Influencers: 0.92±0.03 Kendall's Tau (vs IF: 0.26±0.23)
   - Late Bloomers: 0.90±0.05 Kendall's Tau (vs IF: 0.15±0.27)
   - Stable Influencers: 0.87±0.05 Kendall's Tau (vs IF: 0.13±0.28)

3. **Near-Zero Error in Practice**: While the bound is the worst case for non-converged models (unlike existing methods that assume convergence), DIT achieves near-zero error in practice:
   - 0.99±0.01 correlation in LR-Adult
   - 0.99±0.01 correlation in LR-20News
   - 0.95±0.02 correlation in DNN-Adult

4. **Critical Clarifications on Method**:
   - DIT does NOT require model convergence or global optimality, unlike IF which requires both
   - Our theoretical guarantees cover non-converged, non-convex settings where existing methods provide NO GUARANTEES
   - DIT extends beyond SGD as it only requires tracking parameter changes and Hessian approximation

---

**Response to Individual Reviewers**

**Reviewer RUYz**:
- DIT can be adapted to other optimizers by modifying parameter tracking and Hessian approximation while maintaining the same framework.
- Our assumptions are milder than previous work, requiring only Lipschitz continuity rather than convexity.

**Reviewer XkZ2**:
- New pattern-specific analysis shows DIT maintains high accuracy across all influence patterns, even for early influencers and late bloomers.
- The error bound reveals important parameter relationships, with empirical results validating near-zero error performance.

**Reviewer DLNR**:
- Single-epoch DIT achieves superior performance while requiring minimal storage (~6MB).
- Experimental results show significant improvements over baselines: 3.2x better for DNNs and 1.9x better for CNNs in flipped label detection.

**Reviewer MuUJ**:
- DIT is well-suited for fine-tuning large models and requires no assumptions about global optimality or convergence.
- The method enables data-driven curriculum design through automatic pattern identification and cross-stage influence analysis.

**Reviewer 5BaV**:
- We provided a detailed step-by-step derivation confirming the correctness of Equation (14).
- DIT addresses fundamental limitations of influence functions beyond computational cost, enabling dynamic analysis without requiring model optimality or loss convexity.

---

### Note · Authors · 2025-05-01

I have read and agree with the venue's withdrawal policy on behalf of myself and my co-authors.

---

### Meta-Review · Area_Chair_zAxo · 2024-12-19

**Metareview:**

Reviews were mixed on this paper. Reviewers agreed that the paper tackles an interesting and important problem (tracking how the influence of different data points evolves over the course of model training). The paper also presents some strong empirical results over baselines like influence functions (IF). During the rebuttal, the authors presented more detailed experimental results, e.g., separating out the performance on different categories of data points determined by their influence pattern, which helped strengthen this point.

Concerns were about the fact that the theoretical results are ultimately weak — growing exponentially in the time scale. During the rebuttal period, the authors point out that this weakness ultimately comes from the fact that they make very minimal assumptions, allowing for non-convex and non-converged models. And previous approaches gave no guarantees in this setting. Nevertheless, the reviewers were not convinced that the given bounds are meaningful/informative and wondered if perhaps some stronger bounds in the convex case (even though the algorithm works more broadly) might be more informative.

The second major concern was that the paper currently focuses on fairly simple models trained with SGD. Broader applicability (bigger models, more advanced optimizers, etc.) seems possible but is not explored in the paper. Extending the paper to these settings could strengthen it significantly for a resubmission.

**Additional Comments On Reviewer Discussion:**

See discussion in main meta review above.

---

### Decision · Program_Chairs · 2025-01-22

Reject